# A National Survey on the Clinical Practice Patterns of Korean Medicine Doctors for Attention Deficit Hyperactivity Disorder (ADHD) in Children and Adolescents

**DOI:** 10.3390/children10091490

**Published:** 2023-08-31

**Authors:** Jihong Lee

**Affiliations:** Department of Korean Pediatrics, College of Korean Medicine, Daegu Haany University, 136 Sinchendong–ro, Suseong-gu, Daegu 42158, Republic of Korea; jihonglee@dhu.ac.kr; Tel.: +82-53-770-2080

**Keywords:** attention deficit hyperactivity disorder, ADHD, child, Korean traditional medicine, herbal medicine, surveys and questionnaires

## Abstract

To alleviate the symptoms of attention deficit/hyperactivity disorder (ADHD) in children and reduce the side effects of psychostimulants, parents are opting for complementary and alternative medicine as a therapeutic option. Korean medicine (KM) has been used by Korean medicine doctors (KMDs) to treat ADHD with herbal medication and acupuncture. This study aimed to conduct a cross-sectional survey on the clinical practice patterns of KM for ADHD in children targeting KMDs. The questionnaire included aspects related to patient characteristics, diagnosis, treatment modalities, and perceptions regarding KM. Questionnaires were distributed to all KMDs via e-mail, and the online surveys were conducted from 1 to 15 February 2023. A total 2.1% of KMDs (*n* = 537/25,574) completed the survey. The predominant diagnostic pattern identification employed was “depressed liver qi transforming into fire” (19.8%). Herbal medicine (HM) was the most common treatment (44.2%). The most frequently used HM prescriptions were Ondam-tang (16.9%), Eokgan-san (15.7%), and Sihogayonggolmoryeo-tang (14.4%). KMDs recognized HM as the most effective among the KM treatments (54.3%). The results of this study elucidate the current clinical practice patterns of KM for ADHD. Based on these findings, a treatment protocol can be developed to provide optimized KM treatment services to patients with ADHD.

## 1. Introduction

Attention deficit/hyperactivity disorder (ADHD) is one of the most prevalent neurodevelopmental disorders in childhood and is characterized by age-inappropriate inattention, hyperactivity, and impulsivity [1,2]. ADHD in children and adolescents often persists into adulthood, disrupting healthy growth and impairing social, academic, and occupational functioning [2]. Its prevalence was estimated to be 7.2% in children in a meta-analysis [3] and 8.4% in a patient-reported national survey of pediatric and adolescent populations in the United States [4]. Its prevalence in the Republic of Korea varies depending on research methodology and geographical region [5,6]. A prevalence rate of 1.99% was reported in a study involving elementary school students screened for ADHD using teacher checklists and child interviews in rural areas of Korea [5]. Furthermore, when elementary school students in a medium-sized Korean city (consisting of urban, rural, and industrial areas) were screened using the Korean version of the ADHD Rating Scale, a screening tool for ADHD, a standardized diagnosis was obtained by pediatric psychiatrists, and the prevalence of pediatric ADHD was found to be 8.5% [6]. The number of ADHD patients in the Republic of Korea has increased by a factor of 1.87 over the past four years from 53,070 in 2017 to 99,488 in 2021 [7].

According to the Health Insurance Review and Assessment Service Big Data Open portal [7], the total medical treatment expenditure for ADHD in the Republic of Korea amounts to 60 billion KRW, which has increased by a factor of 2.4 over the past four years. In the United States, the health- and work-related costs of ADHD amount to $32 billion annually [8]. The economic burden caused by ADHD in children and adolescents includes direct costs, such as medical expenses and indirect costs, such as special education, additional care, and loss of work absence to care for the children [9,10]. The total annual societal excess costs associated with ADHD were estimated to be $19.4 billion for children and $13.8 billion for adolescents in the United States, with direct healthcare costs from claims data and direct non-healthcare and indirect costs from government publications and the literature [10].

In the management of pediatric patients with ADHD, pharmacotherapy and behavioral therapy are commonly employed either separately or in combination [1]. Methylphenidate, a stimulant, is effective in short-term improvement in ADHD symptoms among children and adolescents. However, the long-term effect of this pharmacological treatment is controversial and has received limited investigation [11,12]. In addition, the administration of stimulants may give rise to common side effects, such as decreased appetite, weight loss, headache, sleep disturbance, and abdominal pain as well as the less frequent occurrence of hallucinations, elevated heart rate, and heightened blood pressure [1,13]. Therefore, additional measures are needed for children who either fail to experience long-term effects from psychostimulants or are hampered by side effects.

The Republic of Korea’s dual medical system provides both conventional Western medicine (WM) and Korean medicine (KM) modalities [14,15]. KM, rooted in East Asian traditions and practiced in Korea, China, and Japan, employs diagnostic and therapeutic procedures including pattern identification (PI), herbal medicine, acupuncture, and moxibustion. PI entails the comprehensive assessment of a patient’s clinical manifestations through four types of examinations, namely inspection, listening and smelling, inquiry, and palpation, culminating in the derivation of disease causality, nature, and location as well as the patient’s physiological state and a treatment strategy through thorough analysis [16,17]. KM applies a systemic and personalized approach to diagnose and treat diseases [18].

Various complementary and alternative medicines (CAMs) have been used to improve ADHD symptoms and reduce the risk of ADHD by minimizing the use of psychostimulants with interventions, such as behavioral therapy, parental counseling, herbal medicine, and acupuncture [18,19]. East Asian traditional medicine uses herbal medicine [20,21,22] and acupuncture [23,24,25,26] to treat ADHD. In comparison to other CAM therapies, acupuncture is regarded as an economical and simple-to-use treatment for ADHD [23]. The use of acupuncture [23,27] or herbal treatment [20], in addition to conventional treatment, has demonstrated superior clinical efficacy in alleviating ADHD symptoms compared to that of conventional treatment alone. However, because standard treatment protocols and clinical practice guidelines for herbal medicine and acupuncture for ADHD have not been established, Korean medicine doctors (KMDs) find it difficult to use reliable evidence to optimize KM treatment. In addition, among the herbal medicines, only 56 types of herbal prescription preparations are guaranteed by health insurance; meanwhile, herbal decoctions, which are the most common formulations among herbal medicines, are not covered by health insurance [28]. Therefore, statistical data, such as the number of patients treated, treatment duration, and prescription frequency, are warranted. Importantly, to the best of my knowledge, no comprehensive study has thus far addressed the treatment patterns for ADHD in children and adolescents as applied within the field of KM, targeting all KMDs.

This study was aimed at conducting an online survey of the clinical practice patterns of KM, including the clinical characteristics of patients, diagnostic tools, PI, treatment methods, and perceptions of KM treatment for ADHD in children and adolescents, which are currently being performed by KMDs. Although this study has a limitation in its reliance on self-reported data from KMDs, the anticipation is that the results of this study will serve as preliminary data for systematic reviews or the development of treatment protocol in the future.

## 2. Materials and Methods

### 2.1. Study Design

This was a cross-sectional web-based survey targeting KMDs conducted to understand the current status of KM treatment for ADHD.

### 2.2. Questionnaire Development

A questionnaire was developed to investigate the current clinical patterns of KM treatment for ADHD. The initial draft of the questionnaire was constructed by a researcher who is a licensed KMD, a KM pediatrician with more than 15 years of clinical experience, and a professor of pediatrics at KM University. The initial draft was based on a questionnaire from existing survey articles on KM treatment [29,30,31]. By referring to the domain classification and question format of the existing questionnaires, details such as diagnosis tools, terms of PI, and treatment method (herbal medicine prescription and acupoints) were modified to be tailored to the disease. The draft was reviewed for item appropriateness by two external experts, who are professors at different KM Universities and KM pediatricians with more than 15 years of clinical experience. Based on their comments, the draft was revised, and the questionnaire was finalized. The final version of the questionnaire consisted of 38 questions (14 sub-questions) across six domains. The survey items belonging to each domain are as follows:(1)Sociodemographic data of respondents: Sex, age, years of clinical experience, place of work, and type of specialty credentials;(2)Clinical characteristics of ADHD patients: Monthly number of first-time patients, treatment period, cost per treatment, main age group of ADHD patients treated, concurrent Western treatment, and referrals to Western medical institutions;(3)Diagnosis of ADHD: Diagnostic tools used in diagnosis, pattern identification (PI);(4)Treatment of ADHD: Main treatment methods, name of herbal medicine, frequently used single herbs and acupoints, duration of herbal medicine and acupuncture, types of herbal medicine formulation, acupuncture, pharmacopuncture, and manual or exercise therapy;(5)Perceptions of KM treatment for ADHD: Effects of KM treatment for children with ADHD, effective KM treatment methods, advantages and disadvantages of KM treatment, and items that require additional training for KMDs;(6)Evaluation of the safety and effectiveness of KM treatment: Periods and indicators.

Domains (3) and (4) intended to use information on the diagnosis and treatment used by the KMD as reference data for future research of actual patients or systematic reviews. The data obtained through domains (5) and (6) were used for the development and application of treatment protocols for ADHD and training medical personnel.

Questions about PI, name of the herbal medicine prescription, single herbs, and acupoints were developed with reference to KM pediatric textbooks [32]. To ensure that there were no unanswered items, moving to the next domain was not permitted if there were items left unanswered. For cases who stopped without completing the response, the data were excluded from the analysis. The Appendix A shows the full survey.

### 2.3. Distribution and Collection of Questionnaires

In the Republic of Korea, an individual who has completed a 6-year curriculum at a university of KM is eligible to take the KMD’s licensing exam, and licensed KMDs who have passed the exam can choose a specialist course consisting of a 1-year internship and a 3-year residency. Upon successful completion of this course and passing the qualification examination, the individual attains the status of a Korean Medicine specialist. This specialization encompasses eight distinct fields: internal medicine, acupuncture and moxibustion, pediatrics, gynecology, neuropsychiatry, otorhinolaryngology and dermatology, rehabilitation medicine, and Sasang constitutional medicine. As of 2020, among the 22,038 licensed KMDs, 3293 (14.9%) were KM specialists with licenses. Within this cohort of specialists, 198 (6%) were categorized as KM neuropsychiatrists and 121 (3.7%) were designated as KM pediatric specialists. These specialized fields are closely associated with the treatment of conditions like ADHD [33]. The requirements for the survey respondents were as follows: (1) licensed KMDs, (2) members of the Association of Korean Medicine whose e-mail addresses are registered with the association, and (3) those who agreed to participate. Through cooperation with the Association of Korean Medicine, e-mails were sent to all KMD members on 1 February 2023. The email contained the description of the study and the link to the survey. The web version of the survey was prepared using the online survey site Moaform (https://www.moaform.com/, accessed on 20 February 2023). The survey site was set up to allow only one input per Internet Protocol (IP) address assigned to the participant, preventing multiple responses from one computer. For the survey created on this platform, the response data can be downloaded as an Excel file without including information that can identify the respondent. The responses were collected between 1 February 2023 and 15 February 2023. In order to enhance the response rate, reminders were sent, and gift certificates were provided through a lottery.

In the Republic of Korea, an individual who has completed a 6-year curriculum at a university of KM is eligible to take the KMD’s licensing exam, and licensed KMDs who have passed the exam can choose a specialist course consisting of a 1-year internship and a 3-year residency. Upon successful completion of this course and passing the qualification examination, the individual attains the status of a Korean Medicine specialist. This specialization encompasses eight distinct fields: internal medicine, acupuncture and moxibustion, pediatrics, gynecology, neuropsychiatry, otorhinolaryngology and dermatology, rehabilitation medicine, and Sasang constitutional medicine. As of 2020, among the 22,038 licensed KMDs, 3293 (14.9%) were KM specialists with licenses. Within this cohort of specialists, 198 (6%) were categorized as KM neuropsychiatrists and 121 (3.7%) were designated as KM pediatric specialists. These specialized fields are closely associated with the treatment of conditions like ADHD [33]. The requirements for the survey respondents were as follows: (1) licensed KMDs, (2) members of the Association of Korean Medicine whose e-mail addresses are registered with the association, and (3) those who agreed to participate. Through cooperation with the Association of Korean Medicine, e-mails were sent to all KMD members on 1 February 2023. The email contained the description of the study and the link to the survey. The web version of the survey was prepared using the online survey site Moaform (https://www.moaform.com/, accessed on 20 February 2023). The survey site was set up to allow only one input per Internet Protocol (IP) address assigned to the participant, preventing multiple responses from one computer. For the survey created on this platform, the response data can be downloaded as an Excel file without including information that can identify the respondent. The responses were collected between 1 February 2023 and 15 February 2023. In order to enhance the response rate, reminders were sent, and gift certificates were provided through a lottery.

### 2.4. Ethical Considerations

This study was approved by the Institutional Review Board of Daegu Haany University (IRB No. 2022-4-01). All participants were informed of the purpose and details of the study, and anonymity was guaranteed through an e-mail cover letter prior to participation. Participants were asked to respond to the questionnaire only if they voluntarily agreed to participate and were informed that they could withdraw at any time, even after they had started responding.

### 2.5. Statistical Analyses

Descriptive statistics were used, and categorical data were presented as frequencies and percentages. Continuous variables are expressed as mean ± standard deviation. Multiple-response analysis was used for questions that allowed multiple responses. For categorical data of clinical experience of the KMDs, linear-by-linear association was used, and for categorical data of the level of medical institution, chi-square or Fisher’s exact test were used. A *p*-value < 0.05 was set to determine statistical significance. Data were analyzed using Microsoft Excel 2016 (Microsoft Corporation, Redmond, WA, USA) and SPSS (IBM Corporation, Armonk, NY, USA).

## 3. Results

### 3.1. Sociodemographic Characteristics of Respondents

A total of 25,574 KMDs registered with the Korean Medical Association received an e-mail, and 537 (2.1%) of them checked the e-mail and completed the survey. All of them satisfied the requirements of the participants, and 537 KMDs were included in the final analysis. The respondents’ demographic characteristics are presented in Table 1 and Appendix A. Among the respondents, 375 (69.8%) were male. In terms of age, those in their 40s accounted for the majority with 38.7%, followed by those in their 30s with 32.8%. In terms of clinical experience, 127 KMDs (23.6%) with 10–15 years of clinical experience accounted for the largest proportion, and 411 KMDs (76.5%) worked in Korean medicine clinics, primary healthcare institutions. Among the respondents, 155 (28.9%) were KM specialists doctors who received specialist training.

### 3.2. Current Status of Korean Medicine Treatment for ADHD in Children and Adolescents

The current status of the respondents’ treatment of ADHD in children and adolescents is presented in Table 2. Most KMDs (*n* = 475, 88.5%) reported that the monthly average number of first-time ADHD patients was less than five in the past year. As for the average treatment period, one–three months accounted for 29.6%, followed by less than one month with 27.2%, and three–six months with 25.7%. The main age group of ADHD patients under treatment was children in early elementary school (7–9 years), which accounted for the largest proportion at 47.2%, followed by children in the upper grades of elementary school (10–12 years) at 26%. A total of 24.2% of the KMDs responded that the rate of combining Western treatment for ADHD was 25–50%, followed by 50–75% (23.5%), and 75–100% (22.3%). When ADHD patients received WM treatment, the most common type of treatment was medications (43.7%); 442 respondents (82.3%) stated that the number of direct referrals to WM institutions in the past year was zero, and the main purpose of referral was for diagnosis and examination of ADHD (48.4%), followed by WM prescriptions at 35.8%.

### 3.3. Diagnosis for Children and Adolescents with ADHD

Of the responding KMDs, 74.1% answered that they use PI when diagnosing ADHD, and the most commonly used PI method was “PI of Qi, Blood, Fluid, Humor, and Organ system diagnosis” based on the KM textbooks (62.8%) (Table 3). The title of the mainly used pattern identification was “depressed liver qi transforming into fire” (19.8%), followed by “phlegm-fire harassing the heart” (15.5%), “dual Deficiency of the heart-spleen” (15.4%), and “effulgent heart-liver fire” (15%). Regarding the diagnostic tools, 45.7% of KMDs responded that they assessed by clinical symptoms without using assessment tools, followed by 22.9% of KMDs responding that they used DSM-5, and 17.5% made referrals to other medical institutions or evaluation using records of other medical institutions.

### 3.4. KM Treatments for Children and Adolescents with ADHD

As shown in Table 4, herbal medicine was the most common treatment (44.2%) in KM treatments for children and adolescents with ADHD. Next were acupuncture (29.5%) and moxibustion (8.3%). The mainly used type of herbal medicine formulation was a compound herbal decoction, which accounted for 67.2% of the users. Among the herbal medicine prescriptions, the following were used frequently in the following order: Ondam-tang (16.9%), Eokgan-san (15.7%), Sihogayonggolmoryeo-tang (14.4%), and Gwibi-tang (13.4%). Polygalae Radix (25.5%), Acori Graminei Rhizoma (20.2%), and Poria (18.5%) were frequently used as single herbs. The average treatment period recommended for herbal medicine treatment was more than three months and less than six months, with 44.7% responding the most.

The most commonly used acupuncture was “Meridian points acupuncture” (49.5%). As for acupoints, PC6 (14.9%), GV20 (14.4%), and LI4 (14.3%) were found to be used frequently. The recommended duration of acupuncture was also more than three months and less than six months (36.2%).

The most commonly used pharmacopuncture treatment was Cervi Pantotrichum Cornu (44.9%), followed by Hominis Placenta (26.5%). The most frequently used manual and exercise therapy was Chuna manual therapy (50.9%), followed by acupressure (28.3%).

When the usage pattern of the main KM treatment modalities was analyzed by linear-by-linear association according to clinical experience, the longer the clinical experience, the higher the usage rate of herbal medicines (*p* = 0.037) (Table 5). Acupuncture was not tended to be used based in clinical experience (*p* = 0.262). However, the use of herbal medicine or acupuncture treatment did not differ significantly according to the level of the affiliated medical institution when the chi-square or Fisher’s exact test was used (both *p* > 0.05).

### 3.5. Perception of KM Treatments for Children and Adolescents with ADHD (n = 537)

Based on the treatment experience, the average score was 3.49 ± 0.76 on a five-point Likert scale when asked about the overall effect of KM treatment on children and adolescents with ADHD (1 = not at all effective, 2 = mostly not effective, 3 = moderate, 4 = mostly effective, and 5 = very effective) (Table 6). Most of the respondents (54.3%) thought that herbal medicine treatment was the most effective treatment for children and adolescents with ADHD while 24.8% responded with acupuncture. A total of 33.7% of the respondents considered the fact that Korean medicine treatment for ADHD has fewer side effects an advantage; 30% of the respondents perceived that, in addition to ADHD symptoms, it could improve overall health conditions; and 23.5% said it was effective. As for the aspects that are needed to be supplemented with regards to KM treatment for ADHD, 26.7% of the respondents chose treatment cost, and 23.9% chose lack of treatment information (publicity, awareness). For the areas in which KMDs think additional education is needed regarding ADHD, diagnosis of ADHD accounted for the highest proportion at 28.1%, and use of evaluation tools accounted for 24.3%.

### 3.6. Evaluation of Safety and Effectiveness

Regarding the items on the evaluation period for the effectiveness of ADHD treatment in children and adolescents, 46.6% answered three months, and 36.5% answered one month (Table 7). As for the effectiveness evaluation indicators, 48.6% of the respondents judged through clinical symptoms without indicators, and 14% used ADHD Rating Scale-IV. Regarding the safety evaluation period, one month was the most common (42.5%), followed by three months (40.2%). As for the safety evaluation indicators, 38.7% of respondents checked for changes in the child’s overall condition, and 34.6% checked for adverse events.

## 4. Discussion

The present study was a web-based, cross-sectional survey targeting KMDs on the use of KM treatment for ADHD in children and adolescents. The questionnaire items included information about the patients’ clinical characteristics, diagnosis, treatment, and perceptions. A total of 537 KMDs completed the survey, and 411 (76.5%) of them were KMDs working at KM clinics, which are primary medical institutions. As of 2020, 16,138 (73.2%) of the total number of 22,038 KMDs in Republic of Korea are working at KM clinics [28,34]; it seems that the fact stating that KMDs mainly practice in primary medical institutions is reflected in the characteristics of the respondents.

Of the respondents, 24.2% indicated that 25% to 50% of children with ADHD under treatment received concurrent Western treatment; 23.5% answered “50–75%”, and 22.3% answered “75–100%”. As for the type of concurrent treatment, WM prescriptions were the most common (43.7%). On the other hand, most KMDs (82.3%) responded that there was no case of direct requests to Western medical institutions last year. Therefore, it is possible that the patient and guardian chose both medical institutions out of necessity rather than when a KM institution requested a Western medical institution for consultation. Medical care in Korea is a dual medical system in which patients can choose to receive health insurance treatments in WM and KM. According to the 2022 Korean medicine use survey, parents responded that the most important purpose for their children under the age of 19 to use KM was “disease treatment” (43%), followed by “health promotion” (40.5%) and “growth clinic” (27.6%) [35]. For patients and their guardians who choose both WM and KM simultaneously despite not being directly referred, a survey or interview study is required to determine the reason from their point of view. Several studies [36,37] have conducted cross-sectional surveys targeting parents who use CAM for their children. According to the 2017 National Health Interview Survey (NHIS) conducted in the United States, 19.4% of parents of children with ADHD reported using one or more CAMs for their children [36]. According to the results of the 2012 NHIS, which investigated the reasons for using CAM in children with ADHD, the main reason was that “it was helpful in treating ADHD symptoms when combined with conventional treatments” (60%). In addition, most (72.2%) parents did not disclose their child’s use of CAM to their medical doctor; only 8.1% of patients chose CAM based on the doctor’s recommendations. In an Australian survey targeting parents of children aged 5–17 with ADHD [37], the most common reason for using CAM was to minimize ADHD symptoms (40 of 75 responding families). Moreover, 64% of the respondents indicated that pediatricians were aware of their children’s use of CAM. Since KM and WM are simultaneously applied to children with ADHD in Korea, their effectiveness and safety should be properly identified and recognized by medical staff, and understanding and cooperation among medical personnel is required.

In a meta-analysis of 746 children with ADHD [20], combined therapy involving East Asian herbal medicine (EAHM) alongside conventional WM exhibited a superior clinical efficacy rate when compared to conventional WM alone. The integration of EAHM significantly reduced the scores on all five subscales of the Conner’s Parent Rating Scales compared to methylphenidate alone. Notably, no significant difference was observed in the incidence of adverse events (AEs) between the two groups. The most frequent AEs after taking herbal medicines included mild gastrointestinal symptoms such as loss of appetite, nausea, loose stools, and dry mouth. Overall, the study has methodological limitations owing to the high risk of bias associated with double-blinding and pre-registered protocols. In another systematic review (SR) [27], scrutinizing 34 studies, herbal treatment (including Chinese herbal medicine and patent medicine) demonstrated comparable or enhanced effects relative to methylphenidate. Nevertheless, the lack of high-quality studies posed limitations on the robustness of these findings. The supplementary use of acupuncture alongside conventional medicine showed positive effects in improving symptoms of hyperactivity, impulsivity, learning problems, and behavioral problems compared to conventional medicine alone, and no instances of serious AEs were reported [23]. The AEs after acupuncture were mild, encompassing symptoms such as anorexia, headache, and insomnia. Conversely, improvements in anxiety and psychosomatic scores were not statistically significant when compared to the outcomes of conventional medicine alone. For future research based on actual patient treatment data, it is necessary to evaluate the extent of improvement achieved for each sub-symptom of ADHD as well as for each comorbid symptom. In an SR study [38] that analyzed 3 randomized control trials (RCTs), acupuncture and related treatments (including acupuncture, electroacupuncture, ear acupuncture) were compared with conventional therapies (behavioral therapy, pharmacotherapy). The effects of acupuncture and related treatments were found to be beneficial compared to the control group. However, due to the limited evidence, no clear conclusions can be drawn. In addition, in another SR study [27], involving the analysis of 876 patients across 10 studies, acupuncture showed a significantly higher effectiveness rate than pharmacotherapy (methylphenidate hydrochloride) and demonstrated the ability to lower the hyperactivity score. In view of these results, KM treatments such as herbal medicine and acupuncture can be an alternative for children who cannot continue treatment due to side effects of the pharmacotherapy of WM, and KM treatment can be used in parallel to enhance the effect of existing WM.

In East Asian herbal medicines that included traditional Chinese and Korean medicines, based on the concept of PI, different prescriptions can be administered according to the type of symptoms, even with the same diagnosis. Among the PI methods, the KM textbook-based “Qi, Blood, Fluid, Humor, and Organ system pattern identification” was most frequently used. Among them, the frequently used titles of PI were “depressed liver qi transforming into fire” (19.8%), “phlegm-fire harassing the heart” (15.5%), “dual deficiency of the heart-spleen” (15.4%), “effulgent heart-liver fire” (15%), and “kidney deficiency and liver hyperactivity” (13.6%). In a study on the development of a PI questionnaire for ADHD in KM [17], the four main PI types of ADHD were investigated and produced through a literature review of Korean and Chinese medicine textbooks, specialized books, and expert consultations. These were “kidney deficiency and liver hyperactivity”, “dual deficiency of the heart-spleen”, “phlegm-fire harassing the heart”, and “spleen deficiency and liver effulgence”; through the results of this survey, it was found that this frequent PI was similarly used by clinical KMDs in actual clinical practice.

The respondents answered that herbal medicine treatment was used most frequently in KM, and among the names of the herbal medicine prescriptions, Ondam-tang, Eokgan-san, and Sihogayonggolmororyeo-tang were used to treat ADHD in children and adolescents. Ondam-tang (including modified Ondam-tang, written as Wendan-tang or Wendan decoction in Chinese) has been reported to be effective in treating mental disorders [39,40], including anxiety [41,42], sleep disorders [43], and schizophrenia [44] in East Asian medicine. When prescribing Ondam-tang, different herbs or dosages are used to treat different symptoms, and it is also used by attaching the name of the added herb (ex. Huanglian-Wendan decoction) [44]. Since modified forms are widely used in combination, the questionnaire in this study included a modified form in response to Ondam-tang. Eokgan-san has been prescribed for restlessness and agitation in children and has been reported to improve the behavioral and psychological symptoms of dementia [45,46] and Tourette’s syndrome symptoms in children and adolescents for a short period of time [47]. Sihogayonggolmororyeo-tang (written as Chaihu Jia Longgu Muli decoction in Chinese) has been reported to have an antiepileptic effect in an experimental study on rats [48], and in an SR, it was found to be more effective than conventional medicine in the treatment of post-stroke depression [49] and insomnia [50].

The five most frequently prescribed herbs were Polygalae Radix, Acori Graminei Rhizoma, Poria, Zizyphi Spinosae Semen, and Rehmanniae Radix Preparat. Among the above drugs, four, except for Zizyphi Spinosae Semen, were included in the top 12 frequently used herbs in a literature review [51] that analyzed the prescriptions of 88 traditional Chinese medicines for ADHD treatment. In addition, Polygalae Radix, Acori Graminei Rhizoma, and Rehmanniae Radix Preparat were included in the top five most frequently used herbs in an SR study [20], which analyzed 42 RCTs of herbal medicine treatment for ADHD in children and adolescents. The top five frequently used acupoints in acupuncture were PC6, GV20, LI4, LR3, and HT7. These acupoints were also included in the top ten most frequent acupuncture points in another SR that included 14 RCTs [23], which analyzed the results of acupuncture treatment in 1185 children and adolescents with ADHD. The SR study [27], which included 10 RCTs, showed some discrepancies with this result, with only two (PC6 and GV20) of the most frequent acupoints being the same and the others being LR6, SP6, KI3, and GV24. To identify the effective acupoints, an additional comprehensive literature search, evaluation of the strength of the evidence, and expert consensus are required.

In addition, the items to be supplemented were answered in the following order: treatment cost (26.7%), treatment information (promotion and awareness) (23.9%), and convenience of treatment (herbal medicine and acupuncture) (19%). In the section on Korean medicine treatment status, 32.6% answered 10,000–20,000 KRW as the average cost of treatment (copay) per treatment based on the previous year, followed by a response of 20,000–50,000 KRW at 25%. According to the respondents, the average treatment period was ‘1–3 months’ (29.6%), ‘less than 1 month’ (27.2%), and ‘3–6 months’ (25.7%). Among herbal medicines for treating ADHD, the compound herbal decoction emerged as the most frequently used form. Compound herbal decoctions have the advantage of tailoring the prescription to each individual patient, thereby potentially enhancing treatment effectiveness. However, a problem arises from the fact that these decoctions are not covered by health insurance, leading to potential increases in the financial burden faced by patients and their caregivers. Considering that ADHD requires long-term treatment and follow-up, it is necessary to investigate the economic feasibility of KM treatment. Considering that ADHD requires long-term treatment and follow-up, the economic feasibility of KM treatment should be investigated, and the health insurance coverage should be expanded for herbal medicine treatment. Moreover, additional research is needed on the optimal treatment period or frequency of herbal medicines and acupuncture.

Of the respondents, 48.6% said that the evaluation of ADHD treatment effect was based on clinical symptoms without specific indicators, followed by 14% of those who said that ADHD-RS-IV was used. Of the respondents, 46.6% answered that the treatment effect evaluation was performed every three months. In addition, the majority of respondents said that safety was evaluated with ‘Changes in the child’s general condition’ (38.7%), and the most frequent evaluation cycle was once a month (42.5%). Considering that most KMDs work in primary care institutions, cooperation with other medical personnel should be fostered to ensure a clear evaluation of safety and effectiveness. A standardized evaluation protocol that can be used in primary care institutions should be developed, and medical staff should be systematically trained. To this end, opinion gathering and consensus processes, such as interviews and Delphi surveys with experts in KM treatment for children and adolescents with ADHD, should be conducted.

This study had several limitations. First, only a small number of KMDs responded to the questionnaire. There may have been invalid e-mail addresses in the emails of the Association of Korean Medicine, and this result may not reflect the opinions of all KMDs. In future surveys, it can be considered to use a combination of methods such as sending a questionnaire link through a mobile phone registered with the Association of Korean Medicine or conducting a paper survey at a national refresher education conference for KMDs. In addition, it is important to acknowledge that the outcome of this study serves only as a reference for investigating the clinical practice patterns of KM treatment. Based on this, additional information should be collected through in-depth interviews with KMDs who predominantly engage in ADHD treatment or by conducting research using treatment data from medical institutions. Second, this study used a self-reported survey, and a response bias may have occurred. There may have been recall bias, insincere answers, and exaggeration of treatment advantages. Third, the reliability and validity of the questionnaire used in the study were not assessed. While existing questionnaires that investigated the treatment patterns of KMDs were referred to in the design of the questionnaire, none of these questionnaires had been subject to formal reliability or validity testing [29,30,31]. The preliminary draft of the questionnaire incorporated and modified questions from existing questionnaires, which were subsequently reviewed by an external panel of experts for appropriateness. The external expert panel consisted of KM pediatricians who were experienced in conducting survey studies and did not include a statistician. If additional survey research is conducted, with inclusion of a statistician in the questionnaire development and validation processes, the quality of the questionnaire would improve.

Nevertheless, this study was the first to investigate the actual practice of KM treatment for ADHD in children and adolescents by targeting KMDs. Based on the results of this survey, KMDs can be used as references for the diagnosis and treatment of pediatric patients with ADHD. In the future, to accumulate more rigorous evidence, SRs on the effectiveness and safety of each KM treatment and data accumulation research using actual patient data should be conducted. Furthermore, a treatment and evaluation protocol that reflects the clinical practice pattern of current KM treatment will be developed, and medical personnel training will be implemented to enable a more reliable, systematic, and standardized ADHD treatment.

## 5. Conclusions

This study was the first to show the current clinical practice status of KMDs, including the diagnosis, evaluation, treatment, and awareness of ADHD in children and adolescents. Based on this cross-sectional study, it is necessary to accumulate evidence for the treatment of ADHD in children with KM through various types of research, such as SRs on each KM intervention method, studies using patient-care data in an actual medical setting, and randomized control studies. Based on this, a protocol for effective, safe, and optimized KM treatment should be established and implemented by KM medical staff through professional and systematic training.

## Figures and Tables

**Table 1 children-10-01490-t001:** Sociodemographic characteristics of the respondents (*n* = 537).

Factors		N (%)
Sex		
	Male	375 (69.8)
	Female	162 (30.2)
Age (years)		
	≤29	22 (4.1)
	30–39	176 (32.8)
	40–49	208 (38.7)
	50–59	112 (20.9)
	≥60	19 (3.5)
Years of clinical experience		
	<5	70 (13)
	≥5 to <10	109 (20.3)
	≥10 to <15	127 (23.6)
	≥15 to <20	95 (17.7)
	≥20 to <30	107 (19.9)
	≥30 years	29 (5.4)
Place of work		
	Seoul	170 (31.7)
	Gyeonggi-do	129 (24)
	Daegu	44 (8.2)
	Busan	39 (7.3)
Specialist training		
	No	382 (71.1)
	Yes	155 (28.9)
Specialty area of Korean medicine (if applicable)		
	Internal Korean Medicine	42 (27.1)
	Korean acupuncture and moxibustion medicine	35 (22.6)
	Korean Medicine Neuropsychiatry	19 (12.3)
	Korean Medicine Pediatrics	16 (10.3)
Level of healthcare facility of affiliated institution		
	Korean medicine clinics (primary healthcare institutions)	411 (76.5)
	Korean Medicine Hospital (not a university hospital)	52 (9.7)
	university teaching Korean medicine hospital	36 (6.7)
	convalescent hospital	20 (3.7)

**Table 2 children-10-01490-t002:** Current status of Korean medicine treatment for ADHD in children and adolescents (*n* = 537).

Factors		N (%)
Monthly average number of first-time patients with ADHD in children and adolescents based on the last year		
	≤5	475 (88.5)
	6–10	37 (6.9)
	11–15	15 (2.8)
	16–20	5 (0.9)
	≥21	5 (0.9)
Average treatment period for ADHD patients in children and adolescents who visited the hospital based on the last year		
	<1 month	146 (27.2)
	≥1 month to <3 months	159 (29.6)
	≥3 month to <6 months	138 (25.7)
	≥6 month to <1 year	69 (12.8)
	≥1 year to <3 years	22 (4.1)
	≥3 years	3 (0.6)
Average cost of treatment (copay) per treatment for children and adolescents with ADHD based on the last year (KRW) (If decoction is included, it is calculated by dividing the total cost by the number of days of treatment days)		
	<5000	26 (4.8)
	≥5000 to <10,000	66 (12.3)
	≥10,000 to <20,000	175 (32.6)
	≥20,000 to <50,000	134 (25)
	≥50,000 to <100,000	62 (11.5)
	≥100,000	74 (13.8)
Main age group of children and adolescents with ADHD receiving treatment based on the last year ^1^		
	Preschool children (1 or more and less than 7 years)	131 (15.3)
	Children in early elementary school (7–9 years)	406 (47.2)
	Children in the upper grades of elementary school (10–12 years)	224 (26)
	Middle school students (13–15 years)	69 (8)
	High-school students and above (≥16 years)	30 (3.5)
Based on the previous year, proportion of concurrent Western treatment for children and adolescents with ADHD (%)		
	0	64 (11.9)
	≥1 to <25	97 (18.1)
	≥25 to <50	130 (24.2)
	≥50 to <75	126 (23.5)
	≥75 to <100	120 (22.3)
When children and adolescents with ADHD receive concurrent Western treatment, the type of treatment mainly performed ^1,2^		
	Medication (e.g., methylphenidate, atomoxetine, etc.)	422 (43.7)
	Cognitive Behavioral Therapy	188 (19.5)
	Parent and family counselling	149 (15.4)
	Educational interventions (behavior therapy in schools)	80 (8.3)
	Psychotherapy	48 (5)
Number of referrals to Western medical institutions during the treatment of children and adolescents with ADHD in the past year (Including cases of direct request, excluding cases of simply recommending treatment)		
	0	442 (82.3)
	1–5	79 (14.7)
	6–10	13 (2.4)
	11–15	3 (0.6)
Main purpose of referral to Western medical institutions during the treatment of pediatric ADHD patients ^2^ (*n* = 95)		
	For diagnosis and examination of ADHD	46 (48.4)
	To prescribe Western medicine	34 (35.8)
	For psychotherapy	14 (14.7)

ADHD: Attention deficit hyperactivity disorder; ^1^ multiple responses allowed, up to 3 corresponding; ^2^ if applicable.

**Table 3 children-10-01490-t003:** Diagnosis of ADHD in children and adolescents (*n* = 537).

Factors		N (%)
Use of PI for diagnosis		
	Yes.	398 (74.1)
	No. I treat by the prescriptions commonly used for the disease without using PI	139 (25.9)
The main diagnostic method of PI ^1^ (*n* = 398)		
	PI of Qi, Blood, Fluid, Humor, and Organ system diagnosis based on KM textbooks	250 (62.8)
	PI of Sasang constitutional diagnosis	87 (21.9)
	Six-Meridian PI	47 (11.8)
	Other	14 (3.5)
Mainly used title of PI ^1,2^ (*n* = 612)		
	depressed liver qi transforming into fire	121 (19.8)
	phlegm-fire harassing the heart	95 (15.5)
	dual deficiency of the heart-spleen	94 (15.4)
	effulgent heart-liver fire	92 (15)
	kidney deficiency and liver hyperactivity	83 (13.6)
	dual deficiency of the heart-kidney	47 (7.7)
	spleen deficiency and liver effulgence	40 (6.5)
	heart yin deficiency	22 (3.6)
	essence-blood deficiency	14 (2.3)
	internal obstruction of static blood	4 (0.7)
Diagnostic tool ^2^ (*n* = 759)		
	Assessed through clinical symptoms without using diagnostic tools.	347 (45.7)
	DSM-5	174 (22.9)
	Evaluation through referral to other medical institutions or medical records of other medical institutions	133 (17.5)
	DSM-IV	97 (12.8)
	Other	9 (1.2)

PI: pattern identification; KM: Korean medicine; ^1^ if applicable; ^2^ multiple responses allowed, up to 3 corresponding.

**Table 4 children-10-01490-t004:** Frequently used KM treatments for children and adolescents with ADHD (*n* = 537).

Factors		N (%)
Mainly used KM treatment method for children and adolescents with ADHD ^1^ (*n* = 1176)		
	Herbal medicine	521 (44.2)
	Acupuncture	348 (29.5)
	Moxibustion	98 (8.3)
	Electroacupuncture	56 (4.8)
	Manipulation/Exercise therapy	53 (4.5)
	Pharmacopuncture	38 (3.2)
	Cupping therapy	37 (3.1)
	Others	28 (2.4)
Mainly used types of herbal medicine formulations ^1,2^ (*n* = 739)		
	Compound herbal decoction	497 (67.2)
	Soft extract covered by insurance	50 (6.8)
	Mixture of soluble granules covered by insurance	44 (6)
	Mixture of soluble granules not covered by insurance	38 (5.1)
	Pill preparation	32 (4.3)
	Distillation of the compound herbal decoctions	32 (4.3)
	Soft extract not covered by insurance	29 (3.9)
	Powder preparation	18 (2.4)
Frequently used name of the herbal medicine prescription ^2^ (multiple responses allowed, up to five corresponding, *n* = 1565)		
	Ondam-tang (including Gamiondam-tang and Hwangryeonondam-tang)	265 (16.9)
	Eokgan-san (including Eokgan-san-gabanhajinpi)	246 (15.7)
	Sihogayonggolmoryeo-tang	226 (14.4)
	Gwibi-tang (including Gamigwibi-tang)	210 (13.4)
	Gammaekdaejo-tang	130 (8.3)
	Gamisoyo-san	115 (7.3)
	Yukmijihwang-tang	91 (5.8)
	Gyejigayonggolmoryeo-tang	71 (4.5)
	Bojungikgi-tang	50 (3.2)
	Others	46 (2.9)
	Bosimgeonbi-tang	30 (1.9)
	Hwangryeonhaedok-tang	30 (1.9)
Frequently used single herb ^1,2^ (*n* = 1285)		
	Polygalae Radix	328 (25.5)
	Acori Graminei Rhizoma	260 (20.2)
	Poria	238 (18.5)
	Zizyphi Spinosae Semen	203 (15.8)
	Rehmanniae Radix Preparat	121 (9.4)
Recommended average duration of herbal medicine treatment ^2^ (*n* = 520)		
	<1 month	26 (5)
	≥1 month to <3 months	168 (32.2)
	≥3 months to <6 months	233 (44.7)
	≥6 months to <1 year	67 (12.9)
	≥1 year to <3 year	23 (4.4)
	≥3 years	4 (0.8)
Most commonly used acupuncture method ^1,2^ (*n* = 536)		
	Meridian points acupuncture	267 (49.5)
	Sa-am acupuncture therapy	92 (17.1)
	Auricular acupuncture	42 (7.8)
	Five element acupuncture	40 (7.4)
Mainly used acupoints ^1,2^ (*n* = 1276)		
	PC6	191 (14.9)
	GV20	184 (14.4)
	LI4	183 (14.3)
	LR3	143 (11.2)
	HT7	108 (8.4)
Recommended average duration of acupuncture treatment ^2^ (*n* = 347)		
	<1 month	19 (5.5)
	≥1 month to <3 months	107 (30.7)
	≥3 months to <6 months	126 (36.2)
	≥6 months to <1 year	72 (20.7)
	≥1 year to <3 year	18 (5.2)
	≥3 years	6 (1.7)
Mainly used types of pharmacopunctures ^1,2^ (*n* = 49)		
	Cervi Pantotrichum Cornu	22 (44.9)
	Hominis Placenta	13 (26.5)
	Ginseng	7 (14.3)
	Others	7 (14.3)
Mainly used manual/exercise therapy ^2^ (*n* = 53)		
	Chuna manual therapy	27 (50.9)
	acupressure	15 (28.3)
	pediatric tuina	6 (11.3)

KM: Korean medicine; ^1^ multiple responses allowed, up to 3 corresponding; ^2^ if applicable.

**Table 5 children-10-01490-t005:** Use of main treatment modalities by subgroup.

Treatments	Clinical Experience					Level of KM Institution		
	<10 Years (*n* = 179) (*n*, (%))	10–19 Years (*n* = 222) (*n*, (%))	20–29 Years (*n* = 107) (*n*, (%))	≥30 Years (*n* = 29) (*n*, (%))	*p*-Value	Clinic (*n* = 411) (*n*, (%))	Hospital (*n* = 88) (*n*, (%))	*p*-Value
Herbal medicine	170 (95.0)	216 (97.3)	107 (100)	28 (96.6)	0.037 ^a^	398 (96.8)	85 (96.6)	0.557 ^b^
Acupuncture	118 (65.9)	148 (66.7)	67 (62.6)	15 (51.7)	0.262 ^a^	263 (64)	56 (63.6)	0.512 ^c^

KM: Korean medicine. ^a^ linear-by-linear association; ^b^ Fisher’s exact test; ^c^ chi-square test.

**Table 6 children-10-01490-t006:** Perceptions regarding Korean medicine treatments (*n* = 537).

Statements		N (%)
(1) Based on your treatment experience, what is the overall effect of KM treatment on children and adolescents with ADHD?		
	Very effective	44 (8.2)
	Mostly effective	220 (41)
	Moderate	232 (43.2)
	Mostly not effective	39 (7.3)
	Not at all effective	2 (0.4)
(2) Based on your treatment experience, which is the most effective KM treatment for children and adolescents with ADHD? ^1^ (*n* = 930)		
	Herbal medicine	506 (54.3)
	Acupuncture	231 (24.8)
	Moxibustion	54 (5.8)
	Manipulation/Exercise therapy	40 (4.3)
	Pharmacopuncture	35 (3.8)
	Electroacupuncture	33 (3.5)
(3) What do you think are the advantages of KM treatment for children and adolescents with ADHD? ^1^ (*n* = 1153)		
	Fewer side effects	389 (33.7)
	In addition to ADHD symptoms, it can improve a patient’s overall health.	346 (30)
	Effective	271 (23.5)
	More helpful than other treatment methods.	115 (10)
(4) What do you think needs to be supplemented in KM treatment for children and adolescents with ADHD? ^1^ (*n* = 1211)		
	Treatment cost	324 (26.7)
	Treatment information (promotion and awareness)	290 (23.9)
	Convenience of treatment (herbal medicine and acupuncture)	231 (19)
	Diagnostic accuracy	192 (15.8)
	Therapeutic effect	103 (8.5)
(5) What items do you think need additional education for Korean medicine doctors regarding the treatment of ADHD in children and adolescents? ^1^ (*n* = 1210)		
	Diagnosis of ADHD	341 (28.1)
	Utilization of evaluation tools	295 (24.3)
	Details of herbal medicine treatment	230 (19)
	Utilization of psychotherapy	210 (17.3)

KM: Korean medicine; ADHD: Attention deficit hyperactivity disorder; ^1^ multiple responses allowed, up to 3 corresponding.

**Table 7 children-10-01490-t007:** Evaluation of Safety and Effectiveness of KM treatments for ADHD in children and adolescents (*n* = 537).

Factors		N (%)
Evaluation period of the effectiveness in the treatment for ADHD in children and adolescents		
	1 month	196 (36.5)
	3 month	250 (46.6)
	6 month	70 (13)
	1 year	18 (3.4)
Effectiveness evaluation indicators in the treatment of ADHD in children and adolescents ^1^ (*n* = 835)		
	Judgment based on clinical symptoms without specific indicators	407 (48.6)
	ADHD Rating Scale-IV (ADHD-RS-IV)	117 (14)
	Numerical Rating Scale (NRS)	95 (11.4)
	Academic Performance Rating Scales (APRS)	51 (6.1)
	Attention Diagnosis System (ADS)	38 (4.5)
	Comprehensive Attention Test (CAT)	36 (4.3)
	Conners Parents Rating Scale-Revised (CPRS-R)	23 (2.8)
	Brown ADD Rating Scale (Brown ADD-RS)	12 (1.4)
Evaluation period of the safety in the treatment for ADHD in children and adolescents		
	1 month	228 (42.5)
	3 month	216 (40.2)
	6 month	71 (13.2)
	1 year	15 (2.8)
Safety evaluation indicators in the treatment of ADHD in children and adolescents ^1^ (*n* = 1095)		
	Changes in the child’s general condition	425 (38.7)
	Evaluation of Adverse Reactions	380 (34.6)
	Change in vital signs	225 (20.5)
	Blood test	47 (4.3)
	Urine test	17 (1.6)

ADHD: Attention deficit/hyperactivity disorder; ^1^ multiple responses allowed, up to 3 corresponding.

## Data Availability

All the data are presented within the manuscript and the Appendix A. The data presented in this study are available upon request from the corresponding author.

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
