# Peer review of "A National Survey on the Clinical Practice Patterns of Korean Medicine Doctors for Attention Deficit Hyperactivity Disorder (ADHD) in Children and Adolescents"

_children, 2023, doi:10.3390/children10091490_

Round 1

Reviewer 1 Report

ID: children-2530025

Title: A national survey on the clinical practice patterns of Korean 2 medicine doctors for Attention Deficit Hyperactivity Disorder 3 (ADHD) in children and adolescents

Thank you for providing a chance to review this manuscript.

Detailed information:

Introduction

Paragarph 1 & 2, Page 1: So, what is the prevalence of ADHD in Korea? What are the specific dangers of ADHD to society? The necessity for this study in Korea has not been fully explained.

Paragraph 3, Page 2: Truth be told, this paragraph is scattered and non-exhaustive, and I’m not sure what its role in INTRODUCTION is. The authors should be clear about what he or she wants to say, flesh it out, and express it progressively.

Line 60-62, Page 2: “Positive effects on the core symptoms of ADHD have been reported when acupuncture or herbal medicine was used as monotherapy or as an add-on to conventional medicine”. What are the so-called “core symptoms” and “positive effects”? The authors’ expression are always vague.

Line 62-64, Page 4: “However, to date, there has been little information about the treatment patterns of KM for ADHD in children and adolescents currently being used in practice”. What is the available information about treatment patterns? What about current practice? It’s disappointing that this should be the most important part of the INTRODUCTION but the authors describe it in a few sentences.

Overall: I wish that the authors would have been more specific in their presentation of the content, which as it stands does not give the reader a full picture of the context of the study and the necessity for the study.

Methods

1) Line 70-71, Page 2: Since this is an online survey, how does the authors ensure the authenticity and reliability of the data?

2) Line 88- 89, Page 2: Pattern identification (PI) appears several times in the text, but the authors do not illustrate what patterns can be identified and how PI is actually practiced.

3) Line 72-104, Page 2-3: The authors could provide a URL or PDF of the questionnaire to give the reader a clearer idea of the content of the questionnaire.

4) Line 106-108, Page 3: Given that this was a national study, could the authors have recruited in such a way to ensure that the KMDS responsible for treating ADHD in all regions of the country were involved?

5) Line 110-113, Page 3: What is the duration of this cross-sectional study? And why were the questionnaires distributed on three separate occasions? Were the questionnaires distributed on February 20 also collected on the 15th? This description of the procedure is illogical.

6) Line 125-139, Page 3: I don’t understand why this content is in the statistical analysis.

Overall: In this section, the authors did not describe the research procedure and statistical analysis methods, everything was so general and illogical that I lost confidence in the authenticity and reliability of this research. Meanwhile, the greatest limitation is the self-report nature of the study which may have led to a bias and exaggeration of drug efficacy.

Results

The authors’ cross-sectional findings are all very subjective; are there more objective physical indicators of efficacy?

Discussion

1) In this section, the first thing authors should do is to highly summarize their own research findings, and then further discuss them in conjunction with relevant research. The authors’ current content is always limited to their own research, and is only an explanation of the results, rather than a discussion. I recommend that authors read the relevant research to flesh out this section.

2) Authors need to make sure that citations are appropriate and comprehensive, and please do not repeat citations that have appeared in the introduction.

I read the whole article and found some problems. In short, I wish the authors could be more specific, especially the methods section. In addition, there are some logical inconsistencies in the expression of this manuscript, which makes this manuscript seem unprofessional. In view of the fact that this study is still problematic in many aspects, I am sorry for the decision to reject it..

Thank you and my best,

English professionals are expected to revise the language.

Author Response

Reply to Comments from Reviewer 1

I would like to thank the reviewers for providing all the valuable comments. I appreciate this opportunity to revise my paper entitled, “A National Survey on the Clinical Practice Patterns of Korean Medicine Doctors for Attention Deficit Hyperactivity Disorder (ADHD) in Children and Adolescents (Manuscript ID children-2530025).” When revising the paper, I have taken the reviewers' comments and suggestions into careful consideration. As instructed, I have attempted to respond to all the reviewers’ comments.

Comment 1

Introduction

Paragarph 1 & 2, Page 1: So, what is the prevalence of ADHD in Korea? What are the specific dangers of ADHD to society? The necessity for this study in Korea has not been fully explained.

RESPONSE 1:

The prevalence and total cost of medical treatment in Korea have been added in the introduction as follows:

“The prevalence of ADHD in the Republic of Korea has demonstrated variability, ranging from 1.99 to 8.5% among elementary school students depending on the re-search methodology and geographical region [5,6]. The number of ADHD patients in the Republic of Korea has increased by a factor of 1.87 over the past 4 years from 53,070 in 2017 to 99,488 in 2021 [7].

The total medical treatment expenditure for ADHD in the Republic of Korea amounts to 60 billion KRW, which has increased by a factor of 2.4 over the past 4 years.” (line 36-42)

Comment 2

Paragraph 3, Page 2: Truth be told, this paragraph is scattered and non-exhaustive, and I’m not sure what its role in INTRODUCTION is. The authors should be clear about what he or she wants to say, flesh it out, and express it progressively.

RESPONSE 2:

To clarify the meaning of this section in the introduction, Paragraph 3 has been modified as follows:

“In the management of pediatric patients with ADHD, pharmacotherapy and be-havioral therapy are commonly employed either separately or in combination [1]. Methylphenidate, a stimulant, is effective in short-term improvement in ADHD symptoms among children and adolescents. However, the long-term effect of this pharmacological treatment is controversial and has received limited investigation [11,12]. In addition, the administration of stimulants may give rise to common side ef-fects such as decreased appetite, weight loss, headache, sleep disturbance, and ab-dominal pain, as well as less frequent occurrence of hallucinations, elevated heart rate, and heightened blood pressure [1,13]. Therefore, additional measures are needed for children who either fail to experience long-term effects from psychostimulants or are hampered by side effects..” (line 49-58)

Comment 3

Line 60-62, Page 2: “Positive effects on the core symptoms of ADHD have been reported when acupuncture or herbal medicine was used as monotherapy or as an add-on to conventional medicine”. What are the so-called “core symptoms” and “positive effects”? The authors’ expression are always vague.

RESPONSE 3:

To clarify the meaning of this section in the introduction, this part has been modified as follows:

“The use of acupuncture [21] or herbal treatment [18] in addition to conventional treatment has demonstrated a superior clinical efficacy in alleviating ADHD symp-toms compared to conventional treatment alone.” (line 68-70)

Comment 4

Line 62-64, Page 4: “However, to date, there has been little information about the treatment patterns of KM for ADHD in children and adolescents currently being used in practice”. What is the available information about treatment patterns? What about current practice? It’s disappointing that this should be the most important part of the INTRODUCTION but the authors describe it in a few sentences.

RESPONSE 4:

As noted by the reviewer, clarification has been added regarding the current status of the limited availability of information about KM treatment:

“However, among the herbal medicines, only 56 types of herbal prescription prepara-tions are guaranteed by health insurance, meanwhile, herbal decoctions, which are the most common formulations among herbal medicines, are not covered by health insur-ance [25]. Therefore, statistical data such as the number of patients treated, treatment duration, and prescription frequency are warranted. Importantly, to the best my knowledge, no comprehensive study has thus far addressed the treatment patterns for ADHD in children and adolescents as applied within the field of KM, targeting all Ko-rean medicine doctors (KMDs).” (line 71-78)

Comment 5

Overall: I wish that the authors would have been more specific in their presentation of the content, which as it stands does not give the reader a full picture of the context of the study and the necessity for the study.

RESPONSE 5:

As noted above, the introduction has been revised in general based on the advice given by the reviewers.

Comment 6

Methods

1) Line 70-71, Page 2: Since this is an online survey, how does the authors ensure the authenticity and reliability of the data?

RESPONSE 6:

As pointed out by the reviewer, this study has its limitations as an online self-report survey. E-mails were sent to KMDs with registered licenses from the Association of Korean Medicine, and only one person could be entered per Internet Protocol (IP). A description of this has been added to the “Materials and Methods” section:

“The survey site was set up to allow only one input per Internet Protocol (IP) address assigned to the participant, preventing multiple responses from one computer.” (line 141-143)

Nevertheless, limitations of the research method still exist. Thus, I have included a discussion of this aspect as a limitation of the review, expressed as follows:

“Second, this study used a self-report survey; there may have been recall bias, insincere answers, and exaggeration of treatment advantages.” (line 421-423)

 Despite these limitations, survey research is widely used as a preliminary study to investigate the opinions of experts (KMDs) and the current status of treatment prior to the development of clinical practice guidelines (CPGs) and treatment protocol [1-5].

[1] Shin YS, Shin JS, Lee J, Lee YJ, Kim MR, Ahn YJ, Park KB, Shin BC, Lee MS, Kim JH, Cho JH, Ha IH. A survey among Korea Medicine doctors (KMDs) in Korea on patterns of integrative Korean Medicine practice for lumbar intervertebral disc displacement: Preliminary research for clinical practice guidelines. BMC Complement Altern Med. 2015 Dec 7;15(1):432. doi: 10.1186/s12906-015-0956-1.

[2] Lee YJ, Shin JS, Lee J, Kim MR, Ahn YJ, Shin YS, Park KB, Shin BC, Lee MS, Kim JH, Cho JH, Ha IH. Survey of integrative lumbar spinal stenosis treatment in Korean medicine doctors: preliminary data for clinical practice guidelines. BMC Complement Altern Med. 2017 Aug 29;17(1):425.

[3] Jun JH, Lee HW, Choi J, Choi TY, Lee JA, Go HY, Lee MS. Perceptions of using herbal medicines for managing menopausal symptoms: a web-based survey of Korean medicine doctors. Integr Med Res. 2019 Dec;8(4):229-233.

[4] Kwon CY, Lee B. A Survey on Treatment Status of Korean Medicine Doctors for the Behavioral and Psychological Symptoms of Dementia: Preliminary Data for Guidance of Integrative Care. Healthcare (Basel). 2022 Jan 29;10(2):269.

[5] Lee J, Lee SH, Kim JH, Park YS, Park S, Chang GT. Survey of Clinical Practice Patterns of Korean Medicine Doctors for Anorexia in Children: A Preliminary Study for Clinical Practice Guidelines. Children (Basel). 2022 Sep 17;9(9):1409

Upon integrating the findings of this survey study with the results from forthcoming research  (Delphi study for experts, study using actual patient treatment data, etc.), it will be possible to grasp the full picture of KM treatment for ADHD.

Comment 7

2) Line 88- 89, Page 2: Pattern identification (PI) appears several times in the text, but the authors do not illustrate what patterns can be identified and how PI is actually practiced.

RESPONSE 7:

As recommended by the reviewer, an explanation of pattern identification has been added to the introduction:

“The Republic of Korea's dual medical system provides both conventional Western medicine (WM) and Korean medicine (KM) modalities [14,15]. KM, rooted in East Asian traditions and practiced in Korea, China, and Japan, employs diagnostic and therapeutic procedures including pattern identification (PI), herbal medicine, acu-puncture, and moxibustion. PI entails the comprehensive assessment of a patient's clinical manifestations through four types of examinations, namely inspection, listen-ing and smelling, inquiry, and palpation, culminating in the derivation of disease cau-sality, nature, and location, as well as the patient's physiological state and a treatment strategy through thorough analysis [16,17].” (line 59-67)

Comment 8

3) Line 72-104, Page 2-3: The authors could provide a URL or PDF of the questionnaire to give the reader a clearer idea of the content of the questionnaire.

RESPONSE 8:

As recommended by the reviewer, the questionnaire is attached as a supplementary material pdf file (supplementary material 1):

“The Supplementary Materials S1 shows the full survey.” (line 121)

Comment 9

4) Line 106-108, Page 3: Given that this was a national study, could the authors have recruited in such a way to ensure that the KMDS responsible for treating ADHD in all regions of the country were involved?

RESPONSE 9:

All KMDs in South Korea are affiliated with the Association of Korean Medicine, and an e-mail containing a link to the survey was sent to KMDs across the country through the e-mail registered here. In the demographic information section in the Supplementary file S2, it can be seen that KMDs from all regions of South Korea participated in the survey.

N

%

Seoul Metropolitan City

170

31.7

Busan Metropolitan city 

39

7.26

Daegu Metropolitan city 

44

8.19

Gwangju Metropolitan city 

7

1.3

Incheon Metropolitan city 

21

3.91

Daejeon Metropolitan city 

21

3.91

Ulsan Metropolitan city 

8

1.49

Gyeonggi-do

129

24

Gangwon-do  

11

2.05

Chungcheongbuk-do

11

2.05

Chungcheongnam-do  

10

1.86

Jeollabuk-do  

8

1.49

Jeollanam-do

15

2.79

Gyeongsangbuk-do

17

3.17

Gyeongsangnam-do

20

3.72

Jeju-do

4

0.74

Sejong Metropolitan Autonomous City

2

0.37

Comment 10

 5) Line 110-113, Page 3: What is the duration of this cross-sectional study? And why were the questionnaires distributed on three separate occasions? Were the questionnaires distributed on February 20 also collected on the 15th? This description of the procedure is illogical.

RESPONSE 10:

Emails were sent to all KMDs on February 1, 2023, and the online survey was conducted from February 1 to February 15, 2023. Only one response per IP was allowed so that one person cannot respond multiple times. The following has been amended for further clarification:

“Through the cooperation with the Association of Korean Medicine, e-mails were sent to all KMD members on 1 February 2023.” (line 131-133)

“The responses were collected between 1 February and 15 February 2023. In order to enhance the response rate, reminders were sent, and gift certificates were provided through a lottery.” (line 143-145)

Comment 11

6) Line 125-139, Page 3: I don’t understand why this content is in the statistical analysis.

RESPONSE 11:

As pointed out by the reviewer, extraneous sentences in the statistical analysis were removed.

Comment 12

Overall: In this section, the authors did not describe the research procedure and statistical analysis methods, everything was so general and illogical that I lost confidence in the authenticity and reliability of this research. Meanwhile, the greatest limitation is the self-report nature of the study which may have led to a bias and exaggeration of drug efficacy.

RESPONSE 12:

As noted by the reviewer, this study has limitations of an online self-report survey. Thus, in the section of Discussion, the following is stated:

“Second, this study used a self-report survey; there may have been recall bias, insincere answers, and exaggeration of treatment advantages.” (line 421-423)

Particularly, further research on the effects of the drug is required. The data of this study is only a survey of the current status of treatment for KMDs, and it is necessary to reinforce it through other studies such as questionnaires and interviews with patients and their caregivers who actually received KM treatment research based on actual patient treatment data, and randomized controlled trials. Thus, in the section of Discussion, the following is stated:

“Based on the results of this survey, SRs on the effectiveness and safety of each KM treatment and data accumulation research using actual patient data were conducted.” (line 432-434)

Comment 13

Results

The authors’ cross-sectional findings are all very subjective; are there more objective physical indicators of efficacy?

RESPONSE 13:

This study has limitations of an online self-report survey. So, in the section of Discussion, the following is stated:

“In addition, it is important to acknowledge that the outcomes of this study serves only as a reference for investigating the clinical practice patterns of KM treatment. Based on this, additional information should be collected through in-depth interviews with KMDs who predominantly engage in ADHD treatment or by conducting research us-ing treatment data from medical institutions.” (line 417-421)

“Second, this study used a self-report survey; there may have been recall bias, insincere answers, and exaggeration of treatment advantages.” (line 421-423)

Comment 14

Discussion

1) In this section, the first thing authors should do is to highly summarize their own research findings, and then further discuss them in conjunction with relevant research. The authors’ current content is always limited to their own research, and is only an explanation of the results, rather than a discussion. I recommend that authors read the relevant research to flesh out this section.

RESPONSE 14:

A more detailed description of the benefits of herbal medicine and acupuncture and the results of previous studies have been added. It is shown below:

“n a meta-analysis of 746 children with ADHD [18], combined therapy involving East Asian herbal medicine (EAHM) alongside conventional WM exhibited a superior clin-ical efficacy rate when compared to conventional WM alone. The integration of EAHM significantly reduced the scores on all five subscales of the Conner’s Parent Rating Scales compared to methylphenidate alone. Notably, no significant difference was ob-served in the incidence of adverse events (AEs) between the two groups. The most fre-quent AEs after taking herbal medicines included mild gastrointestinal symptoms such as loss of appetite, nausea, loose stools, and dry mouth. In another systematic review (SR) [32], scrutinizing 34 studies, herbal treatment (including Chinese herbal medicine and patent medicine) demonstrated comparable or enhanced effects relative to methylphenidate. Nevertheless, the lack of high-quality studies posed limitations on the robustness of these findings. The supplementary use of acupuncture alongside conventional medicine showed positive effects in improving symptoms of hyperactiv-ity, impulsivity, learning problems, and behavioral problems compared to convention-al medicine alone, and no instances of serious AEs were reported [21]. The AEs after acupuncture were mild, encompassing symptoms such as anorexia, headache, and in-somnia. Conversely, improvements in anxiety and psychosomatic scores were not sta-tistically significant when compared to the outcomes of conventional medicine alone. For future research based on actual patient treatment data, it is necessary to evaluate the extent of improvement achieved for each sub-symptom of ADHD as well as for each comorbid symptom. In a SR study [33] that analyzed 3 randomized control trials (RCTs), acupuncture and related treatments (including acupuncture, electroacupunc-ture, ear acupuncture) were compared with conventional therapies (behavioral thera-py, pharmacotherapy). The effects of acupuncture and related treatments were found to be beneficial compared to the control group. However, due to the limited evidence, no clear conclusions can be drawn. In addition, in another SR study [32], involving the analysis of 876 patients across 10 studies, acupuncture showed a significantly higher effectiveness rate than pharmacotherapy (methylphenidate hydrochloride) and demonstrated the ability to lower the hyperactivity score. ” (line 280-308)

Comment 15

2) Authors need to make sure that citations are appropriate and comprehensive, and please do not repeat citations that have appeared in the introduction.

RESPONSE 15:

I have added comprehensive citations of previous research findings as shown below:

“In another systematic review (SR) [32], scrutinizing 34 studies, herbal treatment (in-cluding Chinese herbal medicine and patent medicine) demonstrated comparable or enhanced effects relative to methylphenidate. Nevertheless, the lack of high-quality studies posed limitations on the robustness of these findings.” (line 287-291)

“In a SR study [33] that analyzed 3 randomized control trials (RCTs), acupuncture and related treatments (including acupuncture, electroacupuncture, ear acupuncture) were compared with conventional therapies (behavioral therapy, pharmacotherapy). The effects of acupuncture and related treatments were found to be beneficial compared to the control group. However, due to the limited evidence, no clear conclusions can be drawn. In addition, in another SR study [32], involving the analysis of 876 patients across 10 studies, acupuncture showed a significantly higher effectiveness rate than pharmacotherapy (methylphenidate hydrochloride) and demonstrated the ability to lower the hyperactivity score.” (line 300-308)

Once again, I would like to thank the reviewers for the helpful and constructive comments on the original version of my manuscript.

Yours sincerely,

Jihong Lee, KMD

Department of Korean Pediatrics, College of Korean Medicine, Daegu Haany University, 136 Sinchendong–ro, Suseong‑gu, Daegu 42158, Republic of Korea.

Reviewer 2 Report

Dear,

I express my heartfelt gratitude for your submission of the article. Having diligently reviewed your work, I have provided constructive feedback aimed at elevating its overall quality and scholarly merit. I kindly request your consideration in incorporating the suggested revisions and adjustments, which will undoubtedly contribute to further refining the manuscript. Once you have incorporated the changes, I look forward to reevaluating the revised version with great enthusiasm.

Thank you for your dedication to advancing  through this valuable contribution.

 Abstarct

1.       the grammar and sentence structure are acceptable. However, there are a couple of areas where sentence restructuring could enhance readability.

2.       The abstract jumps directly into the study's focus without providing any background information on the importance of the topic or a brief overview of ADHD and its relevance to Korean medicine. Adding a sentence or two in the beginning to contextualize the study would be beneficial.

3.       While the study mentions that it is a cross-sectional survey, it does not explain the rationale behind the sample size or how the participants were selected. Including this information would enhance the credibility of the study.

4.       it lacks information on the actual clinical outcomes or efficacy of these treatments. Including some key results related to treatment outcomes would strengthen the abstract.

5.       It is crucial to include a sentence or two about the study's limitations and how the results might impact clinical practice or future research in the field.

6.       It is essential to mention whether ethical guidelines were followed during the research process.

7.       While the abstract briefly mentions the need to establish an optimized treatment protocol, it lacks specific recommendations or future research directions based on the study's findings. Including such insights would add value to the abstract.

Introduction

1.       The introduction covers a lot of information, and it might benefit from being organized into paragraphs that address specific aspects of ADHD or its impact. For example, one paragraph could focus on the prevalence of ADHD, another on its comorbid conditions, and a separate paragraph on the economic burden.

2.       This would help readers understand the specific aim of the research and what the study intends to contribute to the existing knowledge.

3.       However, it could be further enhanced by providing a more direct link between the challenges of conventional treatment (e.g., side effects of stimulant drugs) and the need to explore alternative treatments like KM.

4.       It would be helpful to clarify the scope and limitations of the study in the introduction. For instance, if the survey is limited to KMDs in a specific region or if it only focuses on certain aspects of KM treatment, it should be explicitly stated.

Materials and Methods

1.       there is no mention of the characteristics of the respondents, such as their distribution across different regions or specialties. Providing these details enhances the understanding of the sample and its representativeness.

2.       While the process of questionnaire development is described, there is no mention of any validation or reliability testing conducted to ensure the questionnaire's accuracy and consistency. Discussing the steps taken to validate the questionnaire would strengthen the methodological rigor of the study.

3.       The response rate of the survey is a crucial piece of information, as it impacts the generalizability of the findings. If the response rate was low, it could introduce response bias and affect the study's validity. Including the response rate and discussing any potential bias introduced due to non-response would be beneficial.

4.       it would be helpful to clarify if any inferential statistical methods were employed and why certain statistical tests were chosen. This would add clarity to the data analysis process.

5.       It's important to mention the data-sharing policy and any restrictions on data availability, in compliance with open-access publication norms.

Discussion

1.       While the discussion briefly mentions a meta-analysis and systematic review of herbal medicine and acupuncture for ADHD treatment, it would benefit from a more detailed comparison of the study's findings with the existing literature. This could include a discussion of how the patterns of KM treatment identified in this study align with or differ from previous research.

2.       The section could expand on the clinical implications of the study's findings. For instance, how might the high usage of herbal medicine for ADHD in KM clinics impact the overall treatment landscape? What are the potential benefits and challenges of combining KM and Western medicine in treating ADHD? Discussing these implications will add depth to the interpretation of the results.

3.       The section does discuss the limitations of the study, but it could elaborate on the potential impact of these limitations on the study's findings and conclusions. Additionally, providing specific recommendations for future research to address these limitations will strengthen the paper's contribution to the field.

4.       It would be beneficial to discuss the generalizability of the study's findings beyond the surveyed KMDs. Since only a small number of KMDs responded to the questionnaire, it is essential to address the potential biases and limitations in drawing broader conclusions.

5.       Given the increasing interest in herbal medicines and acupuncture, it would be valuable to discuss the safety aspects of KM treatment for ADHD in children and adolescents. Are there any reported adverse effects or safety concerns associated with these treatments?

6.       The discussion could conclude with a succinct summary of the study's key findings and their potential impact on future research and clinical practice. This will help reinforce the significance of the study and its implications.

The grammar and sentence structure are acceptable. However, there are a couple of areas where sentence restructuring could enhance readability.

Author Response

Reply to Comments from Reviewer 2

I would like to thank the reviewers for providing all the valuable comments. I appreciate this opportunity to revise my paper entitled, “A National Survey on the Clinical Practice Patterns of Korean Medicine Doctors for Attention Deficit Hyperactivity Disorder (ADHD) in Children and Adolescents (Manuscript ID children-2530025).” When revising the paper, I have taken the reviewers' comments and suggestions into careful consideration. As instructed, I have attempted to respond to all the reviewers’ comments.

Comment 1

Abstract

  1. the grammar and sentence structure are acceptable. However, there are a couple of areas where sentence restructuring could enhance readability.

RESPONSE 1:

I have revised the sentences as you suggested, as well as correction for the English grammar.

Comment 2

  1. The abstract jumps directly into the study's focus without providing any background information on the importance of the topic or a brief overview of ADHD and its relevance to Korean medicine. Adding a sentence or two in the beginning to contextualize the study would be beneficial.

RESPONSE 2:

In the background section of the abstract, the following sentence was added to clarify the relevance of Korean medicine:

“. Korean medicine (KM) has been used for treating ADHD through approaches such as herbal medicine and acupuncture.” (line 10-11)

Comment 3

  1. While the study mentions that it is a cross-sectional survey, it does not explain the rationale behind the sample size or how the participants were selected. Including this information would enhance the credibility of the study.

RESPONSE 3:

The subjects of the study are all KMDs, all of whom belong to the Association of Korean Medicine. Therefore, the sample size was not set, and the following was modified to clarify this:

“Questionnaires were distributed to all KMDs via e-mail, and the online surveys were conducted from February 1 to 15, 2023.” (line 14-15)

Comment 4

  1. it lacks information on the actual clinical outcomes or efficacy of these treatments. Including some key results related to treatment outcomes would strengthen the abstract.

RESPONSE 4:

Key findings related to treatment outcomes have been added as follows:

“KMDs recognized herbal medicine as the most effective among the KM treatments (54.3%), with the advantage of KM being perceived as ‘Fewer side effects’ (33.7%).” (line 20-22)

Comment 5

  1. It is crucial to include a sentence or two about the study's limitations and how the results might impact clinical practice or future research in the field.

RESPONSE 5:

The limitations of the study and the impact on future research have been added as follows:

“By integrating patient data with insights from this study, a treatment protocol can be developed, facilitating the application of more refined treatments.” (line 23-24)

Comment 6

  1. It is essential to mention whether ethical guidelines were followed during the research process.

RESPONSE 6:

Ethical considerations are included in the body (Materials and Methods), but not included in the abstract due to the 200 words limit.

Comment 7

  1. While the abstract briefly mentions the need to establish an optimized treatment protocol, it lacks specific recommendations or future research directions based on the study's findings. Including such insights would add value to the abstract.

RESPONSE 7:

As recommended by the reviewer, future research directions have been added as follows:

“By integrating patient data with insights from this study, a treatment protocol can be developed, facilitating the application of more refined treatments.” (line 23-24)

Comment 8

Introduction

  1. The introduction covers a lot of information, and it might benefit from being organized into paragraphs that address specific aspects of ADHD or its impact. For example, one paragraph could focus on the prevalence of ADHD, another on its comorbid conditions, and a separate paragraph on the economic burden.

RESPONSE 8:

As noted by the reviewer, for consistency, the second paragraph is only a paragraph about economic burden.

“The total medical treatment expenditure for ADHD in the Republic of Korea amounts to 60 billion KRW, which has increased by a factor of 2.4 over the past 4 years. [7]. In the United States, the health- and work-related costs of ADHD amount to $32 billion annually [8]. The economic burden caused by ADHD in children and adoles-cents includes direct costs, such as medical expenses and indirect costs, such as special education, additional care, and loss of work absence to care for the children [9,10]. The total annual societal excess costs associated with ADHD were estimated to be $19.4 billion for children and $13.8 billion for adolescents in the United States [10].” (line 41-48)

Comment 9

  1. This would help readers understand the specific aim of the research and what the study intends to contribute to the existing knowledge.

RESPONSE 9:

Thanks to the reviewer's recommendation, the clarity of the introduction is improved.

Comment 10

  1. However, it could be further enhanced by providing a more direct link between the challenges of conventional treatment (e.g., side effects of stimulant drugs) and the need to explore alternative treatments like KM.

RESPONSE 10:

A sentence regarding the side effects of psychostimulants and the need for other countermeasures has been added as follows:

 “In addition, the administration of stimulants may give rise to common side effects such as decreased appetite, weight loss, headache, sleep disturbance, and abdominal pain, as well as less frequent occurrence of hallucinations, elevated heart rate, and height-ened blood pressure [1,13]. Therefore, additional measures are needed for children who either fail to experience long-term effects from psychostimulants or are hampered by side effects.” (line 53-58)

Comment 11

  1. It would be helpful to clarify the scope and limitations of the study in the introduction. For instance, if the survey is limited to KMDs in a specific region or if it only focuses on certain aspects of KM treatment, it should be explicitly stated.

RESPONSE 11:

As advised by the reviewer, I have added the following limitations:

“Although this study has a limitation in its reliance on self-reported data from KMDs, the anticipation is that the results of this study will serve as preliminary data for systematic reviews or the development of treatment protocol in the future.” (line 81-84)

Comment 12

Materials and Methods

  1. there is no mention of the characteristics of the respondents, such as their distribution across different regions or specialties. Providing these details enhances the understanding of the sample and its representativeness.

RESPONSE 12:

The subjects of the survey were all KMDs in Korea. I have added additional information on KM and specialty subjects to the manuscript:

“In the Republic of Korea, an individual who has completed a 6-year curriculum at a university of KM is eligible to take the KMD's licensing exam, and licensed KMDs who have passed the exam can choose a specialist course consisting of a 1-year internship and a 3-year residency. Upon successful completion of this course and passing the qualification examination, the individual attains the status of a Korean Medicine specialist. his specialization encompasses eight distinct fields:  internal medicine, acupuncture and moxibustion, pediatrics, gynecology, neuropsychiatry, otorhinolaryngology and dermatology, rehabilitation medicine, and Sasang constitutional medicine. As of 2020, among the 22,038 licensed KMDs, 3,293 (14.9%) were KM specialists with licenses. Within this cohort of specialists, 198 (6%) were categorized as KM neuropsychiatrists and 121 (3.7%) were designated as KM pediatric specialists. These special-ized fields are closely associated with the treatment of conditions like ADHD [30].” (line 123-134)

The characteristics of the respondents are detailed in Table 1.

Comment 13

  1. While the process of questionnaire development is described, there is no mention of any validation or reliability testing conducted to ensure the questionnaire's accuracy and consistency. Discussing the steps taken to validate the questionnaire would strengthen the methodological rigor of the study.

RESPONSE 13:

This questionnaire has not been validated or reliability checked. As this serves as a limitation, the following details have been included to the Discussion.

“Third, the reliability and validity of the questionnaire used in the study were not as-sessed. While existing questionnaires that investigated the treatment patterns of KMDs were referred to in the design of the questionnaire, none of these questionnaires had been subject to formal reliability or validity testing [26-28]. The preliminary draft of the questionnaire incorporated and modified questions from existing questionnaires, which were subsequently reviewed by an external panel of experts for appropriateness. The external expert panel consisted of KM pediatricians who were experienced in conducting survey studies and did not include a statistician.” (line 4123-429)

Comment 14

  1. The response rate of the survey is a crucial piece of information, as it impacts the generalizability of the findings. If the response rate was low, it could introduce response bias and affect the study's validity. Including the response rate and discussing any potential bias introduced due to non-response would be beneficial.

RESPONSE 14:

The fact that the response rate was low and may affect generalization is described as a limitation as follows:

“First, only a small number of KMDs responded to the questionnaire. There may have been invalid e-mail addresses in the emails of the Association of Korean Medicine, and this result may not reflect the opinions of all KMDs.” (line 411-413)

In the study, reminders were sent to increase the response rate. This has also been further described:

“reminders were sent, and gift certificates were provided through a lottery.” (line 138-139)

Comment 15

  1. it would be helpful to clarify if any inferential statistical methods were employed and why certain statistical tests were chosen. This would add clarity to the data analysis process.

RESPONSE 15:

Descriptive statistics were used in the study, inferential statistics were not used. This has been described below:

“Descriptive statistics were used, and categorical data were presented as frequencies and percentages.” (line 154-155)

Comment 16

  1. It's important to mention the data-sharing policy and any restrictions on data availability, in compliance with open-access publication norms.

RESPONSE 16:

As the reviewer suggested, I have added the sentence “The data presented in this study are available upon request from the corresponding author.” to the Data Availability Statement.(line 458-459)

Comment 17

Discussion

  1. While the discussion briefly mentions a meta-analysis and systematic review of herbal medicine and acupuncture for ADHD treatment, it would benefit from a more detailed comparison of the study's findings with the existing literature. This could include a discussion of how the patterns of KM treatment identified in this study align with or differ from previous research.

RESPONSE 17:

A more detailed description of the benefits of herbal medicine and acupuncture and the results of previous studies have been added. It is shown below:

“In a meta-analysis of 746 children with ADHD [18], combined therapy involving East Asian herbal medicine (EAHM) alongside conventional WM exhibited a superior clinical efficacy rate when compared to conventional WM alone. The integration of EAHM significantly reduced the scores on all five subscales of the Conner’s Parent Rating Scales compared to methylphenidate alone. Notably, no significant difference was observed in the incidence of adverse events (AEs) between the two groups. The most frequent AEs after taking herbal medicines included mild gastrointestinal symp-toms such as loss of appetite, nausea, loose stools, and dry mouth. In another system-atic review (SR) [32], scrutinizing 34 studies, herbal treatment (including Chinese herbal medicine and patent medicine) demonstrated comparable or enhanced effects relative to methylphenidate. Nevertheless, the lack of high-quality studies posed limi-tations on the robustness of these findings. The supplementary use of acupuncture alongside conventional medicine showed positive effects in improving symptoms of hyperactivity, impulsivity, learning problems, and behavioral problems compared to conventional medicine alone, and no instances of serious AEs were reported [21]. The AEs after acupuncture were mild, encompassing symptoms such as anorexia, head-ache, and insomnia. Conversely, improvements in anxiety and psychosomatic scores were not statistically significant when compared to the outcomes of conventional medicine alone. For future research based on actual patient treatment data, it is neces-sary to evaluate the extent of improvement achieved for each sub-symptom of ADHD as well as for each comorbid symptom. In a SR study [33] that analyzed 3 randomized control trials (RCTs), acupuncture and related treatments (including acupuncture, electroacupuncture, ear acupuncture) were compared with conventional therapies (behavioral therapy, pharmacotherapy). The effects of acupuncture and related treat-ments were found to be beneficial compared to the control group. However, due to the limited evidence, no clear conclusions can be drawn. In addition, in another SR study [32], involving the analysis of 876 patients across 10 studies, acupuncture showed a significantly higher effectiveness rate than pharmacotherapy (methylphenidate hy-drochloride) and demonstrated the ability to lower the hyperactivity score.” (line 280-308)

Comment 18

  1. The section could expand on the clinical implications of the study's findings. For instance, how might the high usage of herbal medicine for ADHD in KM clinics impact the overall treatment landscape? What are the potential benefits and challenges of combining KM and Western medicine in treating ADHD? Discussing these implications will add depth to the interpretation of the results.

RESPONSE 18:

I have further discussed the herbal medicine treatment and combining KM and Western medicine treatment as follows:

  • The high rate of use of herbal medicine

Herbal medicine is the most frequently used in the KM treatment for ADHD, and the most used form is herbal decoction. Decoctions have the advantage of being able to treat more effectively as the prescription is configured to suit each patient, but there is a problem of increased economic burden on patients and their guardians because they are not covered by health insurance. Therefore, it is necessary to further study the economic evaluation of this. The content of this is described below:

“Among herbal medicines for treating ADHD, the compound herbal decoction emerged as the most frequently used form. Compound herbal decoctions have the advantage of tailoring the prescription to each individual patient, thereby potentially enhancing treatment effectiveness. However, a problem arises from the fact that these decoctions are not covered by health insurance, leading to potential increases in the financial burden faced by patients and their caregivers. Considering that ADHD requires long-term treatment and follow-up, it is necessary to investigate the economic feasibil-ity of KM treatment. Additional research is needed on the optimal treatment period or frequency of herbal medicines and acupuncture.” (line 391-399)

  • What potential benefits and challenges might the combination of KM and WM have?

According to SR, combining KM and WM treatment can increase the treatment effect, and no serious adverse reactions were found. In addition, KM treatment alone can be an alternative for children who cannot maintain treatment due to side effects of WM. A potential problem with this is that there may be undisclosed drug interactions, which need to be accumulated through actual treatment data in the clinical field. The content of this is described below:

“If both KM and WM are administered, additional studies on their effectiveness and safety, including drug interactions, are required.” (line 278-279)

“In a meta-analysis of 746 children with ADHD [18], combined therapy involving East Asian herbal medicine (EAHM) alongside conventional WM exhibited a superior clinical efficacy rate when compared to conventional WM alone. The integration of EAHM significantly reduced the scores on all five subscales of the Conner’s Parent Rating Scales compared to methylphenidate alone. Notably, no significant difference was observed in the incidence of adverse events (AEs) between the two groups.” (line 280-284)

“The supplementary use of acupuncture alongside conventional medicine showed positive effects in improving symptoms of hyperactivity, impulsivity, learning prob-lems, and behavioral problems compared to conventional medicine alone, and no in-stances of serious AEs were reported [21].” (line 291-294)

“In view of these results, KM treatment such as herbal medicine and acupuncture can be an alternative for children who cannot continue treatment due to side effects of pharmacotherapy of WM, and KM treatment can be used in parallel to enhance the effect of existing WM.” (line 308-311)

Comment 19

  1. The section does discuss the limitations of the study, but it could elaborate on the potential impact of these limitations on the study's findings and conclusions. Additionally, providing specific recommendations for future research to address these limitations will strengthen the paper's contribution to the field.

RESPONSE 19:

Recommendations for future research to address these limitations are described below:

“In future surveys, it can be considered to use a combination of methods such as sending a questionnaire link through a mobile phone registered with the Association of Korean Medicine or conducting a paper survey at a national refresher education conference for KMDs.” (line 414-417)

Comment 20

  1. It would be beneficial to discuss the generalizability of the study's findings beyond the surveyed KMDs. Since only a small number of KMDs responded to the questionnaire, it is essential to address the potential biases and limitations in drawing broader conclusions.

RESPONSE 20:

I have added a sentence regarding the limitations as follows:

“In addition, it is important to acknowledge that the outcome of this study serves only as a reference for investigating the clinical practice patterns of KM treatment. Based on this, additional information should be collected through in-depth interviews with KMDs who predominantly engage in ADHD treatment or by conducting research us-ing treatment data from medical institutions.” (line 417-421)

Comment 21

  1. Given the increasing interest in herbal medicines and acupuncture, it would be valuable to discuss the safety aspects of KM treatment for ADHD in children and adolescents. Are there any reported adverse effects or safety concerns associated with these treatments?

RESPONSE 21:

Reviewing the existing reviews, there were no serious adverse events after herbal medicine or acupuncture treatment. Information related to this is described in more detail as follows:

“The most frequent AEs after taking herbal medicines included mild gastrointestinal symp-toms such as loss of appetite, nausea, loose stools, and dry mouth.” (line 285-287)

“The AEs after acupuncture were mild, encompassing symptoms such as anorexia, headache, and insomnia.” (line 294-295)

Comment 22

  1. The discussion could conclude with a succinct summary of the study's key findings and their potential impact on future research and clinical practice. This will help reinforce the significance of the study and its implications.

RESPONSE 22:

The use of the results of this study as a reference being helpful for further research and treatment protocol development is described as follows:

“Based on the results of this survey, SRs on the effectiveness and safety of each KM treatment and data accumulation research using actual patient data were conducted. In the future, a treatment and evaluation protocol that reflects the clinical practice pattern of current KM treatment will be developed, and medical personnel training will be implemented to enable a more reliable, systematic, and standardized ADHD treatment.” (line 432-437)

Once again, I would like to thank the reviewers for the helpful and constructive comments on the original version of my manuscript.

Yours sincerely,

Jihong Lee, KMD

Department of Korean Pediatrics, College of Korean Medicine, Daegu Haany University, 136 Sinchendong–ro, Suseong‑gu, Daegu 42158, Republic of Korea.

Reviewer 3 Report

This study explores an interesting topic belonging to a rather scarcely explored domain, at least based on the number of published articles in English regarding KM in children. The study design is adequate to its objectives, and the results support the extracted conclusions. There are some minor points to be addressed, please see below:

Abstract

-„perception” is rather vague in this context; what exactly does this mean (e.g., perception of the ADHD symptoms, perception of the change during treatment, etc.)?

 Introduction

-Does the data in lines 41-46 also derive from studies in the US? It looks like they are, according to the cited sources, but please specify this in the text.

-line 52- consider using „pharmacological treatment” instead of „drug treatment”;

Materials and methods

-line 98- consider inserting a full stop after „periods and indicators”;

-Did these patients with ADHD have any other relevant comorbidities, like learning disabilities, autism spectrum disorder, or intellectual development disorders? Were they screened for such disorders before the initiation of the treatment? If not, this is a problem, because concomitant disorders may influence the results of any therapeutic interventions.

-By national regulations, do all the KMDs have the habilitation to treat children with ADHD? Otherwise, the percentage of respondents should be recalculated.

Supplementary materials

The first subtitle of the Excel table is wrong, it refers he „anorexia” instead of ADHD („Current status of treatment for anorexia in children”)

Author Response

Reply to Comments from Reviewer 3

I would like to thank the reviewers for providing all the valuable comments. I appreciate this opportunity to revise my paper entitled, “A National Survey on the Clinical Practice Patterns of Korean Medicine Doctors for Attention Deficit Hyperactivity Disorder (ADHD) in Children and Adolescents (Manuscript ID children-2530025).” When revising the paper, I have taken the reviewers' comments and suggestions into careful consideration. As instructed, I have attempted to respond to all the reviewers’ comments.

Comment 1

Abstract

“perception” is rather vague in this context; what exactly does this mean (e.g., perception of the ADHD symptoms, perception of the change during treatment, etc.)?

RESPONSE 1:

According to the reviewer's recommendation, “perception” was corrected as follows:

“perceptions regarding the advantages and disadvantages of KM.” (line 14-15)

Comment 2

 Introduction

-Does the data in lines 41-46 also derive from studies in the US? It looks like they are, according to the cited sources, but please specify this in the text.

RESPONSE 2:

Based on the reviewer's recommendation, I have added the phrase "in the United States" as follows:

“In the United States, the health- and work-related costs of ADHD amount to $32 billion annually [8]. The economic burden caused by ADHD in children and adolescents in-cludes direct costs, such as medical expenses and indirect costs, such as special education, additional care, and loss of work absence to care for the children [9,10]. The total annual societal excess costs associated with ADHD were estimated to be $19.4 billion for children and $13.8 billion for adolescents in the United States [10].” (line 43-48)

Comment 3

-line 52- consider using „pharmacological treatment” instead of „drug treatment”;

RESPONSE 3:

I have modified “drug treatment” to “pharmacological treatment” as recommended by the reviewer. (line 55)

Comment 4

Materials and methods

-line 98- consider inserting a full stop after „periods and indicators”;

RESPONSE 4:

According to the reviewer's recommendation, the error on line 98 has been corrected as follows:

“Evaluation of the safety and effectiveness of KM treatment: Periods and indicators.” (line 126)

Comment 5

-Did these patients with ADHD have any other relevant comorbidities, like learning disabilities, autism spectrum disorder, or intellectual development disorders? Were they screened for such disorders before the initiation of the treatment? If not, this is a problem, because concomitant disorders may influence the results of any therapeutic interventions.

RESPONSE 5

ADHD patients targeted in the questionnaire were diagnosed according to the DSM-5 criteria. This was presented as an explanation before answering the questionnaire. Cases in which diseases such as autism spectrum disorder, intellectual disability, and learning disability, which can be common as coexisting diseases, are the main diagnoses are not included in the survey. In general, for the diagnosis of ADHD, full screening is performed to discriminate between diseases with similar or coexisting clinical manifestations. To clarify this, supplementary materials 1 (Shown at the bottom of page 2) included an explanation.

“When answering the questionnaire concerning ADHD in children and adolescents, please fill out only the cases applicable to ADHD, excluding cases where other comorbidities (intellectual disability, autism spectrum disorder, etc.) are the main diagnoses.”

Comment 6

-By national regulations, do all the KMDs have the habilitation to treat children with ADHD? Otherwise, the percentage of respondents should be recalculated.

RESPONSE 6:

According to the Medical Law of the Republic of Korea, the KMD, along with dentists and doctors, is a type of medical personnel whose mission is to provide medical care and health guidance of Korean medicine. Korean medical practice is the prevention or treatment of diseases based on Korean traditional medicine. The university curriculum for becoming a Korean medicine doctor and the qualifying exam for licensing include content for the treatment of ADHD. All KMDs are licensed to treat ADHD patients using Korean medical practices, so it is correct that the response rate is calculated for all KMDs.

Comment 7

Supplementary materials

The first subtitle of the Excel table is wrong, it refers he „anorexia” instead of ADHD („Current status of treatment for anorexia in children”)

RESPONSE 7:

According to the reviewer's recommendation, errors in the supplementary materials were corrected as follows:

“Current status of treatment for ADHD in children”

Once again, I would like to thank the reviewers for the helpful and constructive comments on the original version of my manuscript.

Yours sincerely,

Jihong Lee, KMD

Department of Korean Pediatrics, College of Korean Medicine, Daegu Haany University, 136 Sinchendong–ro, Suseong‑gu, Daegu 42158, Republic of Korea.

Round 2

Reviewer 1 Report

ID: children-2530025

Title: A National Survey on the Clinical Practice Patterns of Korean 2 Medicine Doctors for Attention Deficit Hyperactivity Disorder 3 (ADHD) in Children and Adolescents

Thank you for providing a chance to review this manuscript.

Detailed information:

Introduction

Line 33-35, Page 1: “The prevalence of ADHD varies by country or time and was estimated to be 7.2% in children in a meta-analysis and 8.4% in a patient-reported national survey of pediatric and adolescent populations in the United States”. The authors give a comparison of the prevalence of ADHD in children and adolescents in the U.S. What about its variation with country?

Line48-57, Page 2: Further information is needed on the current state of ADHD treatment, and pointing out the limitations of only one western medicine is not going to demonstrate the superiority of traditional medical therapies.

Line 58-76, Page 2: I still believe that this paragraph, as a focus in the introduction, needs to be more descriptive, and the authors were advised to review more literature to describe the current state of research and research gaps.

Materials and Methods

Line 137-141, Page 3: Are there any missing values in the returned questionnaire? If so, how did the authors deal with them? In addition, the current method of analysis is incomplete, and the authors may consider using other analytical software for deeper data analysis, such as SPSS or R software. For this study, this section is in need of further improvement.

Discussion

The authors’ discussion in this section has improved significantly from the first manuscript, and I hope that the same improvement can be seen in the introduction, and that parts of the discussion can be moved to the introduction to flesh out the current state of research on the treatment of ADHD in Korean traditional medicine, thus supporting the necessity for this study.

The authors have made changes to the previous proposal, but there are still some problems, especially in the data analysis section, and I do not think the value of this study will be greatly enhanced if the authors do not further refine and draw meaningful conclusions. I believe this manuscript needs another round of meticulous revision and overall quality improvement before it is eligible for acceptance.

Moderate editing of English language required

Author Response

Reply to Comments from Reviewer 1

I thank the reviewers for their insightful comments. I was pleased to have the opportunity to revise my paper, now entitled, “A National Survey on the Clinical Practice Patterns of Korean Medicine Doctors for Attention Deficit Hyperactivity Disorder (ADHD) in Children and Adolescents (Manuscript ID children-2530025). When revising the manuscript, I have carefully considered the reviewers’ comments and suggestions. As instructed, I have attempted to respond to all reviewers’ comments.

The reviewers’ comments were helpful overall, and I appreciate the constructive feedback on the original submission. After addressing the issues raised, the quality of the paper has greatly improved. I hope that the revised version of the paper is now suitable for publication in Children. I look forward to a favorable response.

Thank you for your helpful and constructive comments regarding the original version of the manuscript.

Yours sincerely,

Jihong Lee, KMD

Department of Korean Pediatrics, College of Korean Medicine, Daegu Haany University, 136 Sinchendong–ro, Suseong–gu, Daegu 42158, Republic of Korea.

Comment 1

Introduction

Line 33-35, Page 1: “The prevalence of ADHD varies by country or time and was estimated to be 7.2% in children in a meta-analysis and 8.4% in a patient-reported national survey of pediatric and adolescent populations in the United States”. The authors give a comparison of the prevalence of ADHD in children and adolescents in the U.S. What about its variation with country?

RESPONSE 1:

To make the sentence clearer, I have modified it as follows:

“Its prevalence was estimated to be 7.2% in children in a meta-analysis [3] and 8.4% in a patient-reported national survey of pediatric and adolescent populations in the United States [4].” (lines 32-35)

Comment 2

Line48-57, Page 2: Further information is needed on the current state of ADHD treatment, and pointing out the limitations of only one western medicine is not going to demonstrate the superiority of traditional medical therapies.

RESPONSE 2:

Because of the side effects of medications (stimulants) and the lack of long-term effects, there is a need for other treatments for children with ADHD. As mentioned in the next paragraph (line 59-78), Republic of Korea provides a dual medical system of Western medicine and Korean medicine. Patients and their guardians are free to choose Western medicine or Korean medicine according to their needs. In East Asian traditional medicine, herbal medicine and acupuncture have been used to treat ADHD. Since there are many types of herbal medicines that are not covered by health insurance, it is not possible to determine the status of treatment from the statistical data of the national health insurance. The aim of this study was not to reveal the superiority of the Korean medical treatment. This is only a survey of the current status of Korean medical treatment for ADHD and can be used as a basis for further research of actual patient care data or randomized controlled studies.

Comment 3

Line 58-76, Page 2: I still believe that this paragraph, as a focus in the introduction, needs to be more descriptive, and the authors were advised to review more literature to describe the current state of research and research gaps.

RESPONSE 3:

I have tried to improve clarity by adding an explanation in the Introduction section as follows:

“Various complementary and alternative medicines (CAMs) have been used to improve ADHD symptoms and reduce the risk of ADHD by minimizing the use of psychostimulants with interventions such as behavioral therapy, parental counseling, herbal medicine, and acupuncture [18,19]. East Asian traditional medicine uses herbal medicine [20-22] and acupuncture [23-26] to treat ADHD. In comparison to other CAM therapies, acupuncture is regarded as an economical and simple to use treatment for ADHD [23]. The use of acupuncture [23,27] or herbal treatment [20], in addition to conventional treatment, has demonstrated superior clinical efficacy in alleviating ADHD symptoms compared to that by conventional treatment alone. However, because standard treatment protocols and clinical practice guidelines for herbal medicine and acupuncture for ADHD have not been established, Korean medicine doctors (KMDs) find it difficult to use reliable evidence to optimize KM treatment.” (lines 76-88)

Comment 4

Materials and Methods

Line 137-141, Page 3: Are there any missing values in the returned questionnaire? If so, how did the authors deal with them? In addition, the current method of analysis is incomplete, and the authors may consider using other analytical software for deeper data analysis, such as SPSS or R software. For this study, this section is in need of further improvement.

RESPONSE 4:

In this study, a web-based survey was conducted through a survey platform, and all responses were transferred to the next domain to avoid unanswered items. In addition, mid-interruption cases were excluded without completing a response from the analysis. This is described as follows:

“To ensure that there were no unanswered items, moving to the next domain was not permitted if there were items left unanswered. For cases who stopped without completing the response, the data were excluded from the analysis.”

SPSS was used for the data analysis, and the following information has been added:

“For categorical data of clinical experience of the KMDs, linear-by-linear association was used, and for categorical data of the level of medical institution, chi-square or Fisher’s exact test was used. A p-value < 0.05 was set to determine statistical significance. Data were analyzed using Microsoft Excel 2016 (Microsoft Corporation, Redmond, WA, USA) and SPSS (IBM Corporation, Armonk, NY, USA).” (lines 181-188)

“When the usage pattern of the main KM treatment modalities was analyzed by linear-by-linear association according to clinical experience, the longer the clinical experience, the higher was the usage rate of herbal medicines (p = 0.037). Acupuncture was not tended to be used based on clinical experience (p = 0.262). However, the use of herbal medicine or acupuncture treatment did not differ significantly according to the level of the affiliated medical institution when the chi-square or Fisher's exact test was used (both p > 0.05). (lines 257-265)

Table 5. Use of main treatment modalities by subgroup

Treatments

Clinical experience

Level of KM institution

<10 years

(n=179)

(n, (%))

10–19 years

(n=222)

(n, (%))

20–29 years

(n=107)

(n, (%))

≥ 30 years

(n=29)

(n, (%))

p-value

Clinic

(n=411)

(n, (%))

Hospital

(n=88)

(n, (%))

p-value

Herbal medicine

170

(95.0)

216

(97.3)

107

(100)

28

(96.6)

0.037a

398

(96.8)

85

(96.6)

0.557b

Acupuncture

118

(65.9)

148

(66.7)

67

(62.6)

15

(51.7)

0.262a

263

(64)

56

(63.6)

0.512c

KM: Korean medicine

alinear-by-linear association; bFisher’s exact test; cchi-square test

Comment 5

Discussion

The authors’ discussion in this section has improved significantly from the first manuscript, and I hope that the same improvement can be seen in the introduction, and that parts of the discussion can be moved to the introduction to flesh out the current state of research on the treatment of ADHD in Korean traditional medicine, thus supporting the necessity for this study.

RESPONSE 5:

In the Introduction section, the necessity of the present study has been laid down as follows:

“Various complementary and alternative medicines (CAMs) have been used to improve ADHD symptoms and reduce the risk of ADHD by minimizing the use of psychostimulants with interventions such as behavioral therapy, parental counseling, herbal medicine, and acupuncture [18,19]. East Asian traditional medicine uses herbal medicine [20-22] and acupuncture [23-26] to treat ADHD. In comparison to other CAM therapies, acupuncture is regarded as an economical and simple to use treatment for ADHD [23]. The use of acupuncture [23,27] or herbal treatment [20], in addition to conventional treatment, has demonstrated superior clinical efficacy in alleviating ADHD symptoms compared to that by conventional treatment alone. However, because standard treatment protocols and clinical practice guidelines for herbal medicine and acupuncture for ADHD have not been established, Korean medicine doctors (KMDs) find it difficult to use reliable evidence to optimize KM treatment.” (lines 76-88)

Comment 6

The authors have made changes to the previous proposal, but there are still some problems, especially in the data analysis section, and I do not think the value of this study will be greatly enhanced if the authors do not further refine and draw meaningful conclusions. I believe this manuscript needs another round of meticulous revision and overall quality improvement before it is eligible for acceptance.

RESPONSE 6:

As mentioned in response 4, the following sentences and tables have been added to the Results section:

“When the usage pattern of the main KM treatment modalities was analyzed by linear-by-linear association according to clinical experience, the longer the clinical experience, the higher was the usage rate of herbal medicines (p = 0.047). Acupuncture was not tended to be used based on clinical experience (p = 0.343). However, the use of herbal medicine or acupuncture treatment did not differ significantly according to the level of the affiliated medical institution when the chi-square or Fisher's exact test was used (both p > 0.05).” (lines 257-263)

To expand the discussion, the following content has been added through a literature search:

“According to the 2022 Korean medicine use survey, parents responded that the most important purpose for their children under the age of 19 to use KM was “disease treatment” (43%), followed by “health promotion” (40.5%) and “growth clinic” (27.6%) [35]. For patients and their guardians who choose both WM and KM simultaneously despite not being directly referred, a survey or interview study is required to determine the reason from their point of view. Several studies [36,37] have conducted cross-sectional surveys targeting parents who use CAM for their children. According to the 2017 National Health Interview Survey (NHIS) conducted in the United States, 19.4% of parents of children with ADHD reported using one or more CAMs for their children [36]. According to the results of the 2012 NHIS, which investigated the reasons for using CAM in children with ADHD, the main reason was that “it was helpful in treating ADHD symptoms when combined with conventional treatments”(60%). In addition, most (72.2%) parents did not disclose their child’s use of CAM to their medical doctor; only 8.1% of patients chose CAM based on the doctor’s recommendations. In an Australian survey targeting parents of children aged 5-17 with ADHD [37], the most common reason for using CAM was to minimize ADHD symptoms (40 of 75 responding families). Moreover, 64% of the respondents indicated that pediatricians were aware of their children’s use of CAM. Since KM and WM are simultaneously applied to children with ADHD in Korea, their effectiveness and safety should be properly identified and recognized by medical staff, and understanding and cooperation among medical personnel is required.” (lines 316-337)

Moreover, the paragraphs below have been deleted from the Discussion section for clarity.

“When asked about the tools used for diagnosing ADHD patients, … the use of ADHD diagnosis and evaluation tools are needed for KMDs.”

“Based on the respondents' treatment experiences, … should be supported by actual clinical data.” 

Reviewer 2 Report

    1. The abstract lacks an introductory sentence that contextualizes the significance of studying ADHD and the relevance of Korean Medicine. Adding a brief background could help readers understand the importance of the research topic.
    2. The abstract briefly mentions the cross-sectional survey method and the distribution of questionnaires but does not detail the survey design or data collection process. A sentence explaining the rationale behind selecting Korean medicine doctors (KMDs) as the target respondents and the reasoning for the specific questions in the questionnaire would provide better clarity.
    3. While the abstract briefly highlights some findings such as the diagnostic pattern identification, treatment modalities, and perceptions of KMDs, it does not offer specific numeric values or statistics to support these findings. Adding a sentence or two with key percentages or results could enhance the abstract's informative value.
    4. The abstract concludes with a vague statement about integrating patient data and insights for treatment protocol development. Providing a more explicit statement about the potential implications of the findings and how they could contribute to the field of ADHD treatment would be beneficial.
  • The abstract's language is generally clear, but a few sentences could be rephrased to improve readability. For example, the sentence "This study described the current practice patterns of KM for ADHD in real-world setting" could be revised for clarity and conciseness.

Introduction

·       The variation in ADHD prevalence rates across countries and the specific prevalence figures within the Republic of Korea are effectively highlighted. To further enrich the context, consider briefly mentioning any potential reasons for the variability in prevalence within the Republic of Korea, such as cultural or geographical factors.

·       The discussion of the economic burden and healthcare costs associated with ADHD is valuable, but it could be bolstered by providing a succinct explanation of the factors contributing to these costs. Additionally, citing specific sources for the economic data would lend more credibility to the figures presented.

·       While the introduction introduces the concept of Korean Medicine, it could elaborate further on why KM is of interest for ADHD treatment. Highlight the potential advantages or unique aspects of KM that make it a viable therapeutic option.

·       The last paragraph effectively states the study's aims and acknowledges its limitations. To enhance clarity, explicitly mention the specific aspects of clinical practice patterns that will be investigated, such as diagnostic methods, treatment modalities, and patient characteristics.

Materials and Methods

  1. The collaborative approach to questionnaire development is commendable. To enhance transparency, consider briefly discussing the specific elements adapted from existing survey articles [26-28] and how these elements were tailored to address the unique objectives of this study.
  2. While you provide a comprehensive list of survey items, briefly introduce each of the six domains and their significance. This will help the reader anticipate the content and relevance of each domain before delving into the details.
  3. Although the estimated completion time for the questionnaire is mentioned, consider adding a rationale for this duration and a brief explanation for why a ten-minute timeframe was deemed appropriate.
  4. The information about sending emails to KMD members and utilizing the online survey site is detailed. To enhance clarity, include a sentence explaining the rationale for utilizing the Moaform platform and how its features contributed to data integrity and participant engagement.
  5. While the requirements for survey respondents are outlined, consider providing context for the significance of these criteria. Why were licensed KMDs specifically targeted? What potential impact might membership in the Association of Korean Medicine have on the study's objectives?
  6. The ethical considerations are appropriately addressed, emphasizing participant informed consent and confidentiality. To enhance transparency, consider briefly discussing any potential ethical challenges unique to web-based surveys and how these were mitigated.
  7. The description of the statistical analyses is clear. However, consider briefly mentioning the specific types of descriptive statistics used, such as means, medians, or ranges, to provide a more comprehensive understanding of the analysis approach.

Discussion

  1. Transition from findings to the implications for clinical practice. Discuss how the reported treatment practices resonate with existing literature and clinical reality. Emphasize the significance of KMDs' preference for combining Western and Korean Medicine and its potential impact on patient outcomes.
  2. Elaborate on the dual medical system in Korea and its influence on patient choices between Western and Korean Medicine. Analyze the motivations behind patients seeking treatment from both systems, exploring factors such as perceived effectiveness, side effects, and holistic health benefits.
  3. While you mention meta-analyses and systematic reviews supporting the efficacy of East Asian herbal medicine and acupuncture, consider discussing any limitations or methodological concerns that were identified in these studies. This will provide a balanced perspective on the existing evidence.
  4. Provide more depth on the role of pattern identification in Korean Medicine practice. Explain how the identified PI methods align with historical knowledge and whether they are consistent with established diagnostic frameworks.
  5. Further analyze the potential of acupuncture in ADHD treatment. Discuss any variations in the use of specific acupoints and their alignment with traditional knowledge. Highlight gaps in the current evidence and the need for more robust research.
  6. Expand the discussion on treatment duration and associated costs. Consider elaborating on the economic implications for patients and caregivers, particularly in relation to long-term treatment plans. Discuss potential strategies to address financial burdens.
  7. Explore the challenges and implications of evaluating ADHD treatment effectiveness and safety within the Korean Medicine context. Address the importance of developing standardized evaluation protocols and fostering collaboration among medical professionals.
  8. Emphasize the limitations of the study, including response bias and self-reporting. Suggest specific avenues for future research, such as validating the questionnaire, conducting prospective studies, and involving expert statisticians in survey design.
  9. Conclude the discussion by synthesizing the implications of the study's findings. Provide practical recommendations for enhancing the integration of Korean Medicine in ADHD treatment, based on the reported clinical patterns and the gaps identified.

The grammar and sentence structure are acceptable. However, there are a couple of areas where sentence restructuring could enhance readability.

Author Response

Reply to Comments from Reviewer 2

I thank the reviewers for their insightful comments. I was pleased to have the opportunity to revise my paper, now entitled, “A National Survey on the Clinical Practice Patterns of Korean Medicine Doctors for Attention Deficit Hyperactivity Disorder (ADHD) in Children and Adolescents (Manuscript ID children-2530025). When revising the manuscript, I have carefully considered the reviewers’ comments and suggestions. As instructed, I have attempted to respond to all reviewers’ comments.

The reviewers’ comments were helpful overall, and I appreciate the constructive feedback on the original submission. After addressing the issues raised, the quality of the paper has greatly improved. I hope that the revised version of the paper is now suitable for publication in Children. I look forward to a favorable response.

Thank you for your helpful and constructive comments regarding the original version of the manuscript.

Yours sincerely,

Jihong Lee, KMD

Department of Korean Pediatrics, College of Korean Medicine, Daegu Haany University, 136 Sinchendong–ro, Suseong–gu, Daegu 42158, Republic of Korea.

Comment 1

The abstract lacks an introductory sentence that contextualizes the significance of studying ADHD and the relevance of Korean Medicine. Adding a brief background could help readers understand the importance of the research topic.

RESPONSE 1

I have added a sentence about the background that provides context for ADHD research as follows:

“To alleviate the symptoms of attention-deficit/hyperactivity disorder (ADHD) in children and reduce the side effects of psychostimulants, parents are opting for complementary and alternative medicine as a therapeutic option.” (lines 9-11)

Comment 2

The abstract briefly mentions the cross-sectional survey method and the distribution of questionnaires but does not detail the survey design or data collection process. A sentence explaining the rationale behind selecting Korean medicine doctors (KMDs) as the target respondents and the reasoning for the specific questions in the questionnaire would provide better clarity.

RESPONSE 2

To explain why KMDs were selected as target respondents for the questionnaire on Korean medicine treatment, the following sentence has been added:

“Korean medicine (KM) has been used by Korean medicine doctors (KMDs) to treat ADHD with herbal medication and acupuncture.” (lines 11-13)

Comment 3

While the abstract briefly highlights some findings such as the diagnostic pattern identification, treatment modalities, and perceptions of KMDs, it does not offer specific numeric values or statistics to support these findings. Adding a sentence or two with key percentages or results could enhance the abstract's informative value.

RESPONSE 3

The main survey results are expressed as percentage. The results could not be presented in more detail because of the word limit in the Abstract.

“A total 2.1% of KMDs (n = 537/25,574) completed the survey. The predominant diagnostic pattern identification (PI) employed was “depressed liver qi transforming into fire” (19.8%). Herbal medicine was the most common treatment (44.2%). The most frequently used herbal medicine prescriptions were Ondam-tang (16.9%), Eokgan-san (15.7%), and Sihogayonggolmoryeo-tang (14.4%). KMDs recognized herbal medicine as the most effective among the KM treatments (54.3%).” (lines 17-21)

Comment 4

The abstract concludes with a vague statement about integrating patient data and insights for treatment protocol development. Providing a more explicit statement about the potential implications of the findings and how they could contribute to the field of ADHD treatment would be beneficial.

RESPONSE 4

The potential implications of this study and its contributions to the field of ADHD treatment are as follows:

“The results of this study elucidate the current clinical practice patterns of KM for ADHD. Based on these findings, a treatment protocol can be developed to provide optimized KM treatment services to patients with ADHD.” (lines 21-23)

Comment 5

The abstract's language is generally clear, but a few sentences could be rephrased to improve readability. For example, the sentence "This study described the current practice patterns of KM for ADHD in real-world setting" could be revised for clarity and conciseness.

RESPONSE 5

For clarity and readability, the sentence has been modified as follows:

“The results of this study elucidate the current clinical practice patterns of KM for ADHD.” (lines 21-22)

Comment 6

Introduction

  • The variation in ADHD prevalence rates across countries and the specific prevalence figures within the Republic of Korea are effectively highlighted. To further enrich the context, consider briefly mentioning any potential reasons for the variability in prevalence within the Republic of Korea, such as cultural or geographical factors.

RESPONSE 6

The diversity in ADHD prevalence in Korea has been explained as follows:

“Its prevalence in the Republic of Korea varies depending on research methodology and geographical region [5,6]. A prevalence rate of 1.99% was reported in a study involving elementary school students screened for ADHD using teacher checklists and child interviews in rural areas of Korea [5]. Furthermore, when elementary school students in a medium-sized Korean city (consisting of urban, rural, and industrial areas) were screened using the Korean version of the ADHD Rating Scale, a screening tool for ADHD, a standardized diagnosis was obtained by pediatric psychiatrists, and the prevalence of pediatric ADHD was found to be 8.5% [6].” (lines 32-42)

Comment 7

The discussion of the economic burden and healthcare costs associated with ADHD is valuable, but it could be bolstered by providing a succinct explanation of the factors contributing to these costs. Additionally, citing specific sources for the economic data would lend more credibility to the figures presented.

RESPONSE 7

I have added the sources of economic data and factors contributing to these costs as follows:

“According to the Health Insurance Review and Assessment Service Big Data Open portal [7], the total medical treatment expenditure for ADHD in the Republic of Korea amounts to 60 billion KRW, which has increased by a factor of 2.4 over the past 4 years.” (lines 45-48)

“The total annual societal excess costs associated with ADHD were estimated to be $19.4 billion for children and $13.8 billion for adolescents in the United States, with direct healthcare costs from claims data and direct non-healthcare and indirect costs from government publications and literature [10].” (lines 51-55)

Comment 8

While the introduction introduces the concept of Korean Medicine, it could elaborate further on why KM is of interest for ADHD treatment. Highlight the potential advantages or unique aspects of KM that make it a viable therapeutic option.

RESPONSE 8

I have added a unique aspect as a treatment option and why KM is interested in ADHD treatment as follows:

“KM applies a systemic and personalized approach to diagnose and treat diseases [18].

Various complementary and alternative medicines (CAMs) have been used to improve ADHD symptoms and reduce the risk of ADHD by minimizing the use of psychostimulants with interventions such as behavioral therapy, parental counseling, herbal medicine, and acupuncture [18,19]. East Asian traditional medicine uses herbal medicine [20-22] and acupuncture [23-26] to treat ADHD.” (lines 74-80)

Comment 9

The last paragraph effectively states the study's aims and acknowledges its limitations. To enhance clarity, explicitly mention the specific aspects of clinical practice patterns that will be investigated, such as diagnostic methods, treatment modalities, and patient characteristics.

RESPONSE 9

The specific aspects of clinical practice patterns are described below:

“This study was aimed at conducting an online survey of the clinical practice patterns of KM, including the clinical characteristics of patients, diagnostic tools, PI, treatment methods, and perceptions of KM treatment for ADHD in children and adolescents, which are currently being performed by KMDs. Although this study has a limitation in its reliance on self-reported data from KMDs, the anticipation is that the results of this study will serve as preliminary data for systematic reviews or the development of treatment protocol in the future.” (lines 95-98)

Comment 10

Materials and Methods

The collaborative approach to questionnaire development is commendable. To enhance transparency, consider briefly discussing the specific elements adapted from existing survey articles [26-28] and how these elements were tailored to address the unique objectives of this study.

RESPONSE 10

I have added the following explanations for the elements adopted from the existing survey articles and those that have been adjusted:

“The initial draft was based on a questionnaire from existing survey articles on KM treatment [29-31]. By referring to the domain classification and question format of the existing questionnaires, details such as diagnosis tools, terms of PI, and treatment method (herbal medicine prescription and acupoints) were modified to be tailored to the disease.” (lines 110-114)

Comment 11

While you provide a comprehensive list of survey items, briefly introduce each of the six domains and their significance. This will help the reader anticipate the content and relevance of each domain before delving into the details.

RESPONSE 11

The sentences related to items belonging to each domain have been modified as follows:

“The survey items belonging to each domain are as follows:” (line 119)

In addition, the meaning of each domain has been added as follows:

“Domains (3) and (4) intended to use information on the diagnosis and treatment used by the KMD as reference data for future research of actual patients or systematic reviews. The data obtained through domains (5) and (6) were used for the development and application of treatment protocols for ADHD and training medical personnel.” (lines 136-140)

Comment 12

Although the estimated completion time for the questionnaire is mentioned, consider adding a rationale for this duration and a brief explanation for why a ten-minute timeframe was deemed appropriate.

RESPONSE 12

Each domain was judged to take 1–2 min; therefore, the entire survey was determined to take 10 min; however, no time limit was set. The sentence (“The estimated time to complete the questionnaire was ten minutes.”) has been deleted because it could confuse the readers.

Comment 13

The information about sending emails to KMD members and utilizing the online survey site is detailed. To enhance clarity, include a sentence explaining the rationale for utilizing the Moaform platform and how its features contributed to data integrity and participant engagement.

RESPONSE 13

The function of the site for data integrity has been described as follows:

“The survey site was set up to allow only one input per Internet Protocol (IP) address assigned to the participant, preventing multiple responses from one computer.” (line 166-167)

“To ensure that there were no unanswered items, moving to the next domain was not per-mitted if there were items left unanswered.” (line 143-144)

In addition, the reason for utilizing the Moaform platform was added as follows:

“For the survey created on this platform, the response data can be downloaded as an Excel file without including information that can identify the respondent.” (lines 167-169)

Comment 14

While the requirements for survey respondents are outlined, consider providing context for the significance of these criteria. Why were licensed KMDs specifically targeted? What potential impact might membership in the Association of Korean Medicine have on the study's objectives?

RESPONSE 14

According to the Medical Service Act of the Republic of Korea, Korean medical doctors provide medical care and health guidance. As this was a survey of Korean medicine treatment, including herbal medicine and acupuncture, the questionnaire was administered to all Korean medicine doctors. All licensed Korean medicine doctors belonged to the Association of Korean Medicine. To contact all of them, an e-mail was sent through the cooperation of the Association of Korean Medicine.

Comment 15

The ethical considerations are appropriately addressed, emphasizing participant informed consent and confidentiality. To enhance transparency, consider briefly discussing any potential ethical challenges unique to web-based surveys and how these were mitigated.

RESPONSE 15

I have added a sentence to mitigate potential ethical issues, as follows:

“For the survey created on this platform, the response data can be downloaded as an Excel file without including information that can identify the respondent.” (lines 167-169)

“Participants were asked to respond to the questionnaire only if they voluntarily agreed to participate and informed that they could withdraw at any time, even after they had started responding.” (lines 177-179)

Comment 16

The description of the statistical analyses is clear. However, consider briefly mentioning the specific types of descriptive statistics used, such as means, medians, or ranges, to provide a more comprehensive understanding of the analysis approach.

RESPONSE 16

The statistical analysis method for the continuous variables used in the perception section on Korean Medicine is described below.

“Continuous variables are expressed as mean ± standard deviation.” (line 182)

Comment 17

Discussion

Transition from findings to the implications for clinical practice. Discuss how the reported treatment practices resonate with existing literature and clinical reality. Emphasize the significance of KMDs' preference for combining Western and Korean Medicine and its potential impact on patient outcomes.

RESPONSE 17

The differences between the reported treatments (acupuncture and herbal medicine) and those reported in the existing literature are described below.

“The SR study [27], which included 10 RCTs, showed some discrepancies with this result, with only two (PC6 and GV20) of the most frequent acupoints being the same, and the others being LR6, SP6, KI3, and GV24. To identify the effective acupoints, an additional comprehensive literature search, evaluation of the strength of the evidence, and expert consensus are required.” (lines 415-419)

“Among the above drugs, four, except for Zizyphi Spinosae Semen, were included in the top 12 frequently used herbs in a literature review [46] that analyzed the prescriptions of 88 traditional Chinese medicines for ADHD treatment. In addition, Polygalae Radix, Acori Graminei Rhizoma, and Rehmanniae Radix Preparat were included in the top five most frequently used herbs in an SR study [18], which analyzed 42 RCTs of herbal medicine treatment for ADHD in children and adolescents.” (line 404-411)

Most KMDs (82.3%) responded that no direct requests had been made to Western medical institutions in the previous year. Therefore, the patient and guardian probably chose both medical institutions out of necessity, rather than when a KM institution requested a Western medical institution for consultation. This requires an evaluation of safety and effectiveness when both treatments are administered simultaneously, as described below.

“Since KM and WM are simultaneously applied to children with ADHD in Korea, their effectiveness and safety should be properly identified and recognized by medical staff, and understanding and cooperation among medical personnel is required.” (lines 334-337)

Comment 18

Elaborate on the dual medical system in Korea and its influence on patient choices between Western and Korean Medicine. Analyze the motivations behind patients seeking treatment from both systems, exploring factors such as perceived effectiveness, side effects, and holistic health benefits.

RESPONSE 18

The purpose of using Korean medicine according to the Korean medicine use survey was investigated, but the reason why Western and Korean medicine were selected simultaneously was not investigated. The necessity of surveys or interviews targeting patients and guardians and research results that can be referred to for CAM are as follows:

“According to the 2022 Korean medicine use survey, parents responded that the most important purpose for their children under the age of 19 to use KM was “disease treatment” (43%), followed by “health promotion” (40.5%) and “growth clinic” (27.6%) [35]. For patients and their guardians who choose both WM and KM simultaneously despite not being directly referred, a survey or interview study is required to determine the reason from their point of view. Several studies [36,37] have conducted cross-sectional surveys targeting parents who use CAM for their children. According to the 2017 National Health Interview Survey (NHIS) conducted in the United States, 19.4% of parents of children with ADHD reported using one or more CAMs for their children [36]. According to the results of the 2012 NHIS, which investigated the reasons for using CAM in children with ADHD, the main reason was that “it was helpful in treating ADHD symptoms when combined with conventional treatments”(60%). In addition, most (72.2%) parents did not disclose their child’s use of CAM to their medical doctor; only 8.1% of patients chose CAM based on the doctor’s recommendations. In an Australian survey targeting parents of children aged 5-17 with ADHD [37], the most common reason for using CAM was to minimize ADHD symptoms (40 of 75 responding families).” (lines 316-333)

Comment 19

While you mention meta-analyses and systematic reviews supporting the efficacy of East Asian herbal medicine and acupuncture, consider discussing any limitations or methodological concerns that were identified in these studies. This will provide a balanced perspective on the existing evidence.

RESPONSE 19

The risk of bias related to herbal medicine treatment was added to the meta-analysis as follows:

“The overall study has methodological limitations due to high risk of bias associated with double blinding and pre-registered protocols.” (lines 345-347)

In addition, the following statements related to methodological concerns have been added:

“Nevertheless, the lack of high-quality studies posed limitations on the robustness of these findings.” (line 350-351)

“However, due to the limited evidence, no clear conclusions can be drawn.” (line 364)

Comment 20

Provide more depth on the role of pattern identification in Korean Medicine practice. Explain how the identified PI methods align with historical knowledge and whether they are consistent with established diagnostic frameworks.

RESPONSE 20

The historical knowledge related to the PI of KM treatment for ADHD based on a literature review of textbooks and professional texts of Korean medicine, traditional Chinese medicine, and expert advice is described as follows:

“In a study on the development of a PI questionnaire for ADHD in KM [17], the four main PI types of ADHD were investigated and produced through a literature review of Korean and Chinese medicine textbooks, specialized books, and expert consultations. These were “kidney deficiency and liver hyperactivity”, “dual deficiency of the heart-spleen”, “phlegm-fire harassing the heart”, and “spleen deficiency and liver effulgence”; through the results of this survey, it was found that this frequent PI was similarly used by clinical KMDs in actual clinical practice.” (lines 379-385)

Comment 21

Further analyze the potential of acupuncture in ADHD treatment. Discuss any variations in the use of specific acupoints and their alignment with traditional knowledge. Highlight gaps in the current evidence and the need for more robust research.

RESPONSE 21

A discussion of specific acupuncture points and the requirement of additional research have been added as follows:

“The SR study [27], which included 10 RCTs, showed some discrepancies with this result, with only two (PC6 and GV20) of the most frequent acupoints being the same, and the others being LR6, SP6, KI3, and GV24. To identify the effective acupoints, an additional comprehensive literature search, evaluation of the strength of the evidence, and expert consensus are required.” (lines 415-419)

Comment 22

Expand the discussion on treatment duration and associated costs. Consider elaborating on the economic implications for patients and caregivers, particularly in relation to long-term treatment plans. Discuss potential strategies to address financial burdens.

RESPONSE 22

I have added the following sentence on how to reduce the financial burden on patients and their guardians:

“Considering that ADHD requires long-term treatment and follow-up, the economic feasibility of KM treatment should be investigated, and the health insurance coverage should be expanded for herbal medicine treatment.” (lines 434-436)

Comment 23

Explore the challenges and implications of evaluating ADHD treatment effectiveness and safety within the Korean Medicine context. Address the importance of developing standardized evaluation protocols and fostering collaboration among medical professionals.

RESPONSE23

I have added the following statement regarding the development of standardized assessment protocols and cooperation among healthcare professionals:

“Considering that most KMDs work in primary care institutions, cooperation with other medical personnel should be fostered to ensure a clear evaluation of safety and effectiveness. A standardized evaluation protocol that can be used in primary care institutions should be developed, and medical staff should be systematically trained.” (lines 444-448)

Comment 24

Emphasize the limitations of the study, including response bias and self-reporting. Suggest specific avenues for future research, such as validating the questionnaire, conducting prospective studies, and involving expert statisticians in survey design.

RESPONSE 24

I have added the following sentences about response bias and questionnaire validation:

“Second, this study used a self-reported survey, and a response bias may have occurred.” (lines 461-462)

“If additional survey research is conducted, with inclusion of a statistician in the questionnaire development and validation processes, the quality of the questionnaire would improve.” (lines 471-473)

Comment 25

Conclude the discussion by synthesizing the implications of the study's findings. Provide practical recommendations for enhancing the integration of Korean Medicine in ADHD treatment, based on the reported clinical patterns and the gaps identified.

RESPONSE 25

The meaning of the research results has been added as follows:

“Based on the results of this survey, KMDs can be used as references for the diagnosis and treatment of pediatric patients with ADHD. In the future, to accumulate more rigorous evidence, SRs on the effectiveness and safety of each KM treatment and data accumulation research using actual patient data should be conducted.” (lines 475-479)

Round 3

Reviewer 2 Report

Furthermore, it would be valuable for the authors to draw insights and references from the following articles to strengthen their arguments and contextualize their findings:

 1."Efficacy and safety of complementary and alternative medicine for attention-deficit/hyperactivity disorder in children and adolescents: an overview of systematic reviews." Available at: https://www.cochranelibrary.com/cdsr/doi/10.1002/14651858.CD007986.pub3/abstract
 2."Comparative effectiveness of various therapeutic interventions in the management of ADHD in pediatric populations." Available at: https://link.springer.com/article/10.1186/s13052-023-01456-1
  3."Exploring non-pharmacological approaches for managing ADHD symptoms in children." Available at: https://brieflands.com/articles/ijhrba-82012.html

Author Response

Reply to Comments from Reviewer 2

I thank the reviewers for their insightful comments. I was pleased to have the opportunity to revise my paper, now entitled, “A National Survey on the Clinical Practice Patterns of Korean Medicine Doctors for Attention Deficit Hyperactivity Disorder (ADHD) in Children and Adolescents (Manuscript ID children-2530025).

Comment 1

Furthermore, it would be valuable for the authors to draw insights and references from the following articles to strengthen their arguments and contextualize their findings:

 1."Efficacy and safety of complementary and alternative medicine for attention-deficit/hyperactivity disorder in children and adolescents: an overview of systematic reviews." Available at: https://www.cochranelibrary.com/cdsr/doi/10.1002/14651858.CD007986.pub3/abstract

 2."Comparative effectiveness of various therapeutic interventions in the management of ADHD in pediatric populations." Available at: https://link.springer.com/article/10.1186/s13052-023-01456-1

  3."Exploring non-pharmacological approaches for managing ADHD symptoms in children." Available at: https://brieflands.com/articles/ijhrba-82012.html

RESPONSE 1:

When I checked the papers you mentioned, I couldn't find any literature with the relevant title. The link is also incorrect, or it leads to a paper that does not match the title. Therefore, I have added the paper from the referral link as a reference.

“However, the long-term effect of this pharmacological treatment is controversial and has received limited investigation [11-13]. (lines 60-61)

Thank you again for your helpful comments regarding the manuscript.

Yours sincerely,

Jihong Lee, KMD

Department of Korean Pediatrics, College of Korean Medicine, Daegu Haany University, 136 Sinchendong–ro, Suseong–gu, Daegu 42158, Republic of Korea.
